# On the Convergence and Robustness of Batch Normalization

## Abstract

Despite its empirical success, the theoretical underpinnings of the stability, convergence and acceleration properties of batch normalization (BN) remain elusive. In this paper, we attack this problem from a modeling approach, where we perform a thorough theoretical analysis on BN applied to a simplified model: ordinary least squares (OLS). We discover that gradient descent on OLS with BN has interesting properties, including a scaling law, convergence for arbitrary learning rates for the weights, acceleration effects, as well as insensitivity to the choice of learning rates. We then demonstrate numerically that these findings are not specific to the OLS problem and hold qualitatively for more complex supervised learning problems. This points to a new direction towards uncovering the mathematical principles that underlies batch normalization.

## 1    Introduction

Batch normalization (Ioffe & Szegedy, 2015) (BN) is one of the most important techniques for training deep neural networks and has proven extremely effective in avoiding gradient blowups during back-propagation and speeding up convergence. In its original introduction (Ioffe & Szegedy, 2015), the desirable effects of BN are attributed to the so-called "reduction of covariate shift". However, it is unclear what this statement means in precise mathematical terms. To date, there lacks a comprehensive theoretical analysis of the effect of batch normalization.

In this paper, we study the convergence and stability of gradient descent with batch normalization (BNGD) via a modeling approach. More concretely, we consider a simplified supervised learning problem: ordinary least squares regression, and analyze precisely the effect of BNGD when applied to this problem. Much akin to the mathematical modeling of physical processes, the least-squares problem serves as an idealized "model" of the effect of BN for general supervised learning tasks. A key reason for this choice is that the dynamics of GD without BN (hereafter called GD for simplicity) in least-squares regression is completely understood, thus allowing us to isolate and contrast the additional effects of batch normalization.

The modeling approach proceeds in the following steps. First, we derive precise mathematical results on the convergence and stability of BNGD applied to the least-squares problem. In particular, we show that BNGD converges for any constant learning rate $\varepsilon \in (0, 1]$, regardless of the conditioning of the regression problem. This is in stark contrast with GD, where the condition number of the problem adversely affect stability and convergence. Many insights can be distilled from the analysis of the OLS model. For instance, we may attribute the stability of BNGD to an interesting scaling law governing $\varepsilon$ and the initial condition; This scaling law is not present in GD. The preceding analysis also implies that if we are allowed to use different learning rates for the BN rescaling variables ($\varepsilon_a$) and the remaining trainable variables ($\varepsilon$), we may conclude that BNGD on our model converges for any $\varepsilon > 0$ as long as $\varepsilon_a \in (0, 1]$. Furthermore, we discover an acceleration effect of BNGD and moreover, there exist regions of $\varepsilon$ such that the performance of BNGD is insensitive to changes in $\varepsilon$, which help to explain the robustness of BNGD to the choice of learning rates. We reiterate that contrary to many previous works, all the preceding statements are precise mathematical results that we derive for our simplified model.

The last step in our modeling approach is also the most important: we need to demonstrate that these insights are not specific features of our idealized model. Indeed, they should be true characteristics, at least in an approximate sense, of BNGD for general supervised learning problems. We do this

by numerically investigating the convergence, stability and scaling behaviors of BNGD on various datasets and model architectures. We find that the key insights derived from our idealized analysis indeed correspond to practical scenarios.

## 1.1 RELATED WORK

Batch normalization was originally introduced in (Ioffe & Szegedy, 2015) and subsequently studied in further detail in (Ioffe, 2017). Since its introduction, it has become an important practical tool to improve stability and efficiency of training deep neural networks (He et al., 2016; Bottou et al., 2018). Initial heuristic arguments attribute the desirable features of BN to concepts such as "covariate shift", which lacks a concrete mathematical interpretation and alternative explanations have been given (Santurkar et al., 2018). Recent theoretical studies of BN includes (Ma & Klabjan, 2017), where the authors proposed a variant of BN, the diminishing batch normalization (DBN) algorithm and analyzed the convergence of the DBN algorithm, showing that it converges to a stationary point of the loss function. More recently, (Bjorck et al., 2018) demonstrated that the higher learning rates of batch normalization induce a regularizing effect. Another related work is (Kohler et al., 2018), where the authors also considered the convergence properties of BNGD on linear networks (similar to the least-squares problem), as well as other special problems, such as learning halfspaces and extensions. In the OLS case, the authors showed that for a particularly adaptive choice of dynamic learning rate schedule, which can be seen as a fixed effective step size in our terminology (see equation (11) and the discussion that immediately follows), BNGD converges linearly if $\lambda_{max}$ is known. Moreover, the analysis also requires setting the rescaling parameter $a$ every step to satisfy a stationarity condition, instead of simply performing gradient descent on $a$, as is done in the original BNGD.

The present research differs from these previous analysis in an important way - we study the BNGD algorithm itself, and not a special variant. More specifically, we consider constant learning rates (without knowledge of properties of the OLS loss function) and we perform gradient descent on rescaling parameters. We prove that the convergence occurs for even in this case (and in fact, for arbitrarily large learning rates for $\varepsilon$, as long as $0 < \varepsilon_a \leq 1$). This poses more challenges in the analysis and contrasts our work with previous analysis on modified versions of BNGD. This is an important distinction; While a decaying or dynamic learning rate is sometimes used in practice, in the case of BN it is critical to analyze the non-asymptotic, constant learning rate case, precisely because one of the key practical advantages of BN is that a bigger learning rate can be used than that in GD. Hence, it is desirable, as in the results presented in this work, to perform our analysis in this regime.

Finally, through the lens of the least-squares example, BN can be viewed as a type of over-parameterization, where additional parameters, which do not increase model expressivity, are introduced to improve algorithm convergence and stability. In this sense, this is related in effect to the recent analysis of the implicit acceleration effects of over-parameterization on gradient descent (Arora et al., 2018).

## 1.2 ORGANIZATION

Our paper is organized as follows. In Section 2, we outline the ordinary least squares (OLS) problem and present GD and BNGD as alternative means to solve this problem. In Section 3, we demonstrate and analyze the convergence of the BNGD for the OLS model, and in particular contrast the results with the behavior of GD, which is completely known for this model. We also discuss the important insights to BNGD that these results provide us with. We then validate these findings on more general supervised learning problems in Section 4. Finally, we conclude in Section 5.

## 2 BACKGROUND

Consider the simple linear regression model where $x \in \mathbb{R}^d$ is a random input column vector and $y$ is the corresponding output variable. Since batch normalization is applied for each feature separately, in order to gain key insights it is sufficient to the case of $y \in \mathbb{R}$. A noisy linear relationship is assumed between the dependent variable $y$ and the independent variables $x$, i.e. $y = x^T w + noise$

where $w \in \mathbb{R}^d$ is the parameters. Denote the following moments:

$$H := E[xx^T], \quad g := E[xy], \quad c := E[y^2]. \tag{1}$$

To simplify the analysis, we assume the covariance matrix $H$ of $x$ is positive definite and the mean $E[x]$ of $x$ is zero. The eigenvalues of $H$ are denoted as $\lambda_i(H), i = 1, 2, ...d,$. Particularly, the maximum and minimum eigenvalue of $H$ is denoted by $\lambda_{max}$ and $\lambda_{min}$ respectively. The condition number of $H$ is defined as $\kappa := \frac{\lambda_{max}}{\lambda_{min}}$. Note that the positive definiteness of $H$ allows us to define the vector norms $\|.\|_H$ and $\|.\|_{H^{-1}}$ by $\|x\|_H^2 = x^T H x$ and $\|x\|_{H^{-1}}^2 = x^T H^{-1} x$ respectively.

## 2.1 ORDINARY LEAST SQUARES

The ordinary least squares (OLS) method for estimating the unknown parameters $w$ leads to the following optimization problem

$$\min_{w \in \mathbb{R}^d} J_0(w) := \frac{1}{2} E_{x,y}[(y - x^T w)^2] = \frac{c}{2} - g^T w + \frac{1}{2} w^T H w. \tag{2}$$

The gradient of $J_0$ with respect to $w$ is $\nabla_w J_0(w) = Hw - g$, and the unique minimizer is $w = u := H^{-1}g$. The gradient descent (GD) method (with step size or learning rate $\varepsilon$) for solving the optimization problem (2) is given by the iterating sequence,

$$w_{k+1} = w_k - \varepsilon \nabla_w J_0(w_k) = (I - \varepsilon H)w_k + \varepsilon g, \tag{3}$$

which converges if $0 < \varepsilon < \frac{2}{\lambda_{max}} =: \varepsilon_{max}$, and the convergence rate is determined by the spectral radius $\rho_\varepsilon := \rho(I - \varepsilon H) = \max_i\{|1 - \varepsilon\lambda_i(H)|\}$ with

$$\|u - w_{k+1}\| \le \rho_\varepsilon \|u - w_k\|. \tag{4}$$

It is well known (for example see Chapter 4 of (Saad, 2003)) that the optimal learning rate is $\varepsilon_{opt} = \frac{2}{\lambda_{max} + \lambda_{min}}$, where the convergence estimate is related to the condition number $\kappa(H)$:

$$\|u - w_{k+1}\| \le \frac{\kappa - 1}{\kappa + 1} \|u - w_k\|. \tag{5}$$

## 2.2 BATCH NORMALIZATION

Batch normalization is a feature-wise normalization procedure typically applied to the output, which in this case is simply $z = x^T w$. The normalization transform is defined as follows:

$$N_{BN}(z) := \frac{z - E[z]}{\sqrt{\mathrm{Var}[z]}} = \frac{x^T w}{\sigma}, \tag{6}$$

where $\sigma := \sqrt{w^T H w}$. After this rescaling, $N_{BN}(z)$ will be order 1, and hence in order to reintroduce the scale (Ioffe & Szegedy, 2015), we multiply $N_{BN}(z)$ with a rescaling parameter $a$ (Note that the shift parameter can be set zero since $\mathbb{E}[w^T x | w] = 0$). Hence, we get the BN version of the OLS problem (2):

$$\min_{w \in \mathbb{R}^d, a \in \mathbb{R}} J(a, w) := \frac{1}{2} E_{x,y}\left[\left(y - aN_{BN}(x^T w)\right)^2\right] = \frac{c}{2} - \frac{w^T g}{\sigma} a + \frac{1}{2} a^2. \tag{7}$$

The objective function $J(a, w)$ is no longer convex. In fact, it has trivial critical points, $\{(a^*, w^*) | a^* = 0, w^{*T} g = 0\}$, which are saddle points of $J(a, w)$.

We are interested in the nontrivial critical points which satisfy the relations,

$$a^* = \mathrm{sign}(s)\sqrt{u^T H u}, w^* = su, \text{ for some } s \in \mathbb{R} \setminus \{0\}. \tag{8}$$

It is easy to check that the nontrivial critical points are global minimizers, and the Hessian matrix at each critical point is degenerate. Nevertheless, the saddle points are strict (Details can be found in Appendix), which typically simplifies the analysis of gradient descent on non-convex objectives (Lee et al., 2016; Panageas & Piliouras, 2017).

Consider the gradient descent method to solve the problem (7), which we hereafter call batch normalization gradient descent (BNGD). We set the learning rates for $a$ and $w$ to be $\varepsilon_a$ and $\varepsilon$ respectively. These may be different, for reasons which will become clear in the subsequent analysis. We thus have the following discrete-time dynamical system

$$a_{k+1} = a_k + \varepsilon_a \left( \frac{w_k^T g}{\sigma_k} - a_k \right), \tag{9}$$

$$w_{k+1} = w_k + \varepsilon \frac{a_k}{\sigma_k} \left( g - \frac{w_k^T g}{\sigma_k^2} H w_k \right). \tag{10}$$

We now begin a concrete mathematical analysis of the above iteration sequence.

## 3 Mathematical analysis of BNGD on OLS

In this section, we discuss several mathematical results one can derive concretely for BNGD on the OLS problem (7). First, we establish a simple but useful scaling property, which an important ingredient in allowing us to prove a linear convergence result for arbitrary constant learning rates. We also derive the asymptotic properties of the "effective" learning rate of BNGD (to be precisely defined subsequently), which shows some interesting sensitivity behavior of BNGD on the chosen learning rates. Detailed proofs of all results presented here can be found in the Appendix.

### 3.1 Scaling property

In this section, we discuss a straightforward, but useful scaling property that the BNGD iterations possess. Note that the dynamical properties of the BNGD iteration are governed by a set of numbers, or a *configuration* $\{H, u, a_0, w_0, \varepsilon_a, \varepsilon\}$.

**Definition 3.1** (Equivalent configuration). *Two configurations, $\{H, u, a_0, w_0, \varepsilon_a, \varepsilon\}$ and $\{H', u', a_0', w_0', \varepsilon_a', \varepsilon'\}$, are said to be equivalent if for iterates $\{w_k\}$, $\{w_k'\}$ following these configurations respectively, there is an invertible linear transformation $T$ and a nonzero constant $t$ such that $w_k' = Tw_k, a_k' = ta_k$ for all $k$.*

The scaling property ensures that equivalent configurations must converge or diverge together, with the same rate up to a constant multiple. Now, it is easy to check the system has the following scaling law.

**Proposition 3.2** (Scaling property). *Suppose $\mu \neq 0, \gamma \neq 0, r \neq 0, Q^T Q = I$, then*

(1) *The configurations $\{\mu Q^T H Q, \frac{\gamma}{\sqrt{\mu}} Q u, \gamma a_0, \gamma Q w_0, \varepsilon_a, \varepsilon\}$ and $\{H, u, a_0, w_0, \varepsilon_a, \varepsilon\}$ are equivalent.*

(2) *The configurations $\{H, u, a_0, w_0, \varepsilon_a, \varepsilon\}$ and $\{H, u, a_0, rw_0, \varepsilon_a, r^2\varepsilon\}$ are equivalent.*

It is worth noting that the scaling property (2) in Proposition 3.2 originates from the batch-normalization procedure and is independent of the specific structure of the loss function. Hence, it is valid for general problems where BN is used (Lemma A.9). Despite being a simple result, the scaling property is important in determining the dynamics of BNGD, and is useful in our subsequent analysis of its convergence and stability properties (see the sketch of the proof of Theorem 3.3).

### 3.2 Batch normalization converges for arbitrary step size

We have the following convergence result.

**Theorem 3.3** (Convergence for BNGD). *The iteration sequence $(a_k, w_k)$ in equation (9)-(10) converges to a stationary point for any initial value $(a_0, w_0)$ and any $\varepsilon > 0$, as long as $\varepsilon_a \in (0, 1]$. Particularly, we have the following sufficient conditions of converging to global minimizers.*

(1) *If $a_0 w_0^T g > 0$ (or $a_0 = 0, w_0^T g \neq 0$), $\varepsilon_a \in (0, 1]$ and $\varepsilon$ is sufficiently small (the smallness is quantified by Lemma A.13), then $(a_k, w_k)$ converges to a global minimizer.*

(2) *If $\varepsilon_a = 1$ and $\varepsilon > 0$, then $(a_k, w_k)$ converges to global minimizers for almost all initial values $(a_0, w_0)$.*

**Sketch of Proof.**

We first prove that the algorithm converges for any $\varepsilon_a \in (0, 1]$ and small enough $\varepsilon$, with any initial value $(a_0, w_0)$ such that $\|w_0\| \geq 1$ (Lemma A.13). Next, we observe that the sequence $\{\|w_k\|\}$ is monotone increasing, and thus either converges to a finite limit or diverges. The scaling property is then used to exclude the divergent case if $\{\|w_k\|\}$ diverges, then at some $k$ the norm $\|w_k\|$ should be large enough, and by the scaling property, it is equivalent to a case where $\|w_k\|=1$ and $\varepsilon$ is small, which we have proved converges. This shows that $\|w_k\|$ converges to a finite limit, from which the convergence of $w_k$ and the loss function value can be established, after some work. The proof is fully presented in Theorem A.17 and preceding Lemmas.

In addition, using the 'strict saddle point' arguments in (Lee et al., 2016; Panageas & Piliouras, 2017), we can prove the set of initial value for which $(a_k, w_k)$ converges to saddle points has Lebesgue measure 0, provided some conditions, such as when $\varepsilon_a = 1, \varepsilon > 0$ (Lemma A.20). It is important to note that BNGD converges for all step size $\varepsilon > 0$ of $w_k$, independent of the spectral properties of $H$. This is a significant advantage and is in stark contrast with GD, where the step size is limited by $\lambda_{\max}(H)$, and the condition number of $H$ intimately controls the stability and convergence rate. Although we only prove the almost sure convergence to global minimizer for the case of $\varepsilon_a = 1$, we have not encountered convergence to saddles in the OLS experiments even for $\varepsilon_a \in (0, 2)$ with initial values $(a_0, w_0)$ drawn from typical distributions.

### 3.3 CONVERGENCE RATE, ACCELERATION AND ASYMPTOTIC SENSITIVITY

Now, let us consider the convergence rate of BNGD when it converges to a minimizer. Compared with GD, the update coefficient before $Hw_k$ in equation (10) changed from $\varepsilon$ to a complicated term which we named as the *effective* step size or learning rate $\hat{\varepsilon}_k$

$$\hat{\varepsilon}_k := \varepsilon \frac{a_k}{\sigma_k} \frac{w_k^T g}{\sigma_k^2}, \tag{11}$$

and the recurrence relation in place of $u - w_k$ is

$$u - \frac{w_k^T g}{\sigma_k^2} w_{k+1} = (I - \hat{\varepsilon}_k H)\left(u - \frac{w_k^T g}{\sigma_k^2} w_k\right). \tag{12}$$

Consider the dynamics of the residual $e_k := u - (w_k^T g/\sigma_k^2)w_k$, which equals 0 if and only if $w_k$ is a global minimizer. Using the property of $H$-norm (see section A.1), we observe that the effective learning rate $\hat{\varepsilon}_k$ determines the convergence rate of $e_k$ via

$$\|e_{k+1}\|_H \leq \left\|u - \frac{w_k^T g}{\sigma_k^2} w_{k+1}\right\|_H \leq \rho(I - \hat{\varepsilon}_k H)\|e_k\|_H, \tag{13}$$

where $\rho(I - \hat{\varepsilon}_k H)$ is spectral radius of the matrix $I - \hat{\varepsilon}_k H$. The inequality (13) shows that the convergence of $e_k$ (and hence the loss function, see Lemma A.23) is linear provided $\hat{\varepsilon}_k \in (\delta, 2/\lambda_{max} - \delta)$ for some positive number $\delta$. In fact, if we enforce $\hat{\varepsilon}_k = 1/\lambda_{max}$ for each $k$, which is done in the analysis in Kohler et al. (2018), then one immediately obtains the same linear convergence rate. But this requires knowledge of $\lambda_{max}$ (problem-dependent) and a modification the BNGD algorithm. We instead focus our analysis on the original BNGD algorithm.

Next, let us discuss below an acceleration effect of BNGD over GD. When $(a_k, w_k)$ is close to a minimizer, we can approximate the iteration (9)-(10) by a linearized system. The Hessian matrix for BNGD at a minimizer $(a^*, w^*)$ is $\text{diag}(1, H^*/\|w^*\|^2)$, where the matrix $H^*$ is

$$H^* = H - \frac{Huu^T H}{u^T Hu}. \tag{14}$$

The matrix $H^*$ is positive semi-definite ($H^* u = 0$) and has better spectral properties than $H$, such as a lower pseudo-condition number $\kappa^* = \frac{\lambda_{max}^*}{\lambda_{min}^*} \leq \kappa$, where $\lambda_{max}^*$ and $\lambda_{min}^*$ are the maximal and minimal nonzero eigenvalues of $H^*$ respectively. Particularly, $\kappa^* < \kappa$ for almost all $u$ (see section A.1 ). This property leads to acceleration effects of BNGD: When $\|e_k\|_H$ is small, the contraction coefficient $\rho$ in (13) can be improved to a lower coefficient. More precisely, we have the following result:

**Proposition 3.4.** *For any positive number $\delta \in (0, 1)$, if $(a_k, w_k)$ is close to a minimizer, such that $\frac{\lambda_{max}\varepsilon|a_k|}{\sigma_k^2}\|e_k\|_H \leq \delta$, then we have*

$$\|e_{k+1}\|_H \leq \min\{\frac{\rho^*(I - \hat{\varepsilon}_k H^*) + \delta}{1 - \delta}, \rho(I - \hat{\varepsilon}_k H)\}\|e_k\|_H, \tag{15}$$

*where $\rho^*(I - \hat{\varepsilon}_k H) = \max\{|1 - \hat{\varepsilon}_k \lambda_{min}^*|, |1 - \hat{\varepsilon}_k \lambda_{max}^*|\}$.*

Generally, we have $\rho^*(I - \hat{\varepsilon}_k H^*) \leq \rho(I - \hat{\varepsilon}_k H)$ provided $\hat{\varepsilon}_k > 0$, and the optimal rate is $\rho_{opt}^* := \frac{\kappa^* - 1}{\kappa^* + 1} \leq \frac{\kappa - 1}{\kappa + 1} =: \rho_{opt}$, where the inequality is strict for almost all $u$. Hence, the estimate (15) indicates that the optimal BNGD could have a faster convergence rate than the optimal GD, especially when $\kappa^*$ is much smaller than $\kappa$.

Finally, we discuss the dependence of the effective learning rate $\hat{\varepsilon}_k$ (and by extension, the effective convergence rate (13) or (15)) on $\varepsilon$. This is in essence a sensitivity analysis on the performance of BNGD with respect to the choice of learning rate. The explicit dependence of $\hat{\varepsilon}_k$ on $\varepsilon$ is quite complex, but we can nevertheless give the following asymptotic estimates.

**Proposition 3.5.** *Suppose $\varepsilon_a \in (0, 1], a_0 w_0^T g > 0$, and $||g||^2 \geq \frac{w_0^T g}{\sigma_0^2} g^T H w_0$, then*

> *(1) When $\varepsilon$ is small enough, $\varepsilon \ll 1$, the effective step size has a same order with $\varepsilon$, i.e. there are two positive constants, $C_1, C_2$, independent on $\varepsilon$ and $k$, such that $C_1 \leq \hat{\varepsilon}_k/\varepsilon \leq C_2$.*

> *(2) When $\varepsilon$ is large enough, $\varepsilon \gg 1$, the effective step size has order $O(\varepsilon^{-1})$, i.e. there are two positive constants, $C_1, C_2$, independent on $\varepsilon$ and $k$, such that $C_1 \leq \hat{\varepsilon}_k \varepsilon \leq C_2$.*

Observe that for finite $k$, $\hat{\varepsilon}_k$ is a differentiable function of $\varepsilon$. Therefore, the above result implies, via the mean value theorem, the existence of some $\varepsilon_0 > 0$ such that $d\hat{\varepsilon}_k/d\varepsilon|_{\varepsilon=\varepsilon_0} = 0$. Consequently, there is at least some small interval of the choice of learning rates $\varepsilon$ where the performance of BNGD is insensitive to this choice. In fact, empirically this is one commonly observed advantage of BNGD over GD, where the former typically allows for a variety of (large) learning rates to be used without adversely affecting performance. The same is not true for GD, where the convergence rate depends sensitively on the choice of learning rate. We will see later in Section 4 that although we only have a local insensitivity result above, the interval of this insensitivity is actually quite large in practice.

## 4 EXPERIMENTS

Let us first summarize our key findings and insights from the analysis of BNGD on the OLS problem.

1. A scaling law governs BNGD, where certain configurations can be deemed equivalent
2. BNGD converges for any learning rate $\varepsilon > 0$, provided that $\varepsilon_a \in (0, 1]$. In particular, different learning rates can be used for the BN variables $(a)$ compared with the remaining trainable variables $(w)$
3. There exists intervals of $\varepsilon$ for which the performance of BNGD is not sensitive to the choice of $\varepsilon$

In the subsequent sections, we first validate numerically these claims on the OLS model, and then show that these insights go beyond the simple OLS model we considered in the theoretical framework. In fact, much of the uncovered properties are observed in general applications of BNGD in deep learning.

### 4.1 EXPERIMENTS ON OLS

Here we test the convergence and stability of BNGD for the OLS model. Consider a diagonal matrix $H = diag(h)$ where $h = (1, ..., \kappa)$ is a increasing sequence. The scaling property (Proposition 3.2) allows us to set the initial value $w_0$ having same 2-norm with $u$, $\|w_0\| = \|u\| = 1$. Of course, one can verify that the scaling property holds strictly in this case.

Figure 1 gives examples of $H$ with different condition numbers $\kappa$. We tested the loss function of BNGD, compared with the optimal GD (i.e. GD with the optimal step size $\varepsilon_{opt}$), in a large range of step sizes $\varepsilon_a$ and $\varepsilon$, and with different initial values of $a_0$. Another quantity we observe is the effective step size $\hat{\varepsilon}_k$ of BN. The results are encoded by four different colors: whether $\hat{\varepsilon}_k$ is close to the optimal step size $\varepsilon_{opt}$, and whether loss of BNGD is less than the optimal GD. The results indicate that the optimal convergence rate of BNGD can be better than GD in some configurations. This acceleration phenomenon is ascribed to the pseudo-condition number of $H^*$ (discard the only zero eigenvalue) being less than $\kappa(H)$. This advantage of BNGD is significant when the (pseudo)-condition number discrepancy between $H$ and $H^*$ is large. However, if this difference is small, the acceleration is imperceptible. This is consistent with our analysis in section 3.3.

Another important observation is a region such that $\hat{\varepsilon}$ is close to $\varepsilon_{opt}$, in other words, BNGD significantly extends the range of 'optimal' step sizes. Consequently, we can choose step sizes in BNGD at greater liberty to obtain almost the same or better convergence rate than the optimal GD. However, the size of this region is inversely dependent on the initial condition $a_0$. Hence, this suggests that small $a_0$ at first steps may improve robustness. On the other hand, small $\varepsilon_a$ will weaken the performance of BN. The phenomenon suggests that improper initialization of the BN parameters weakens the power of BN. This experience is encountered in practice, such as (Cooijmans et al., 2016), where higher initial values of BN parameter are detrimental to the optimization of RNN models.

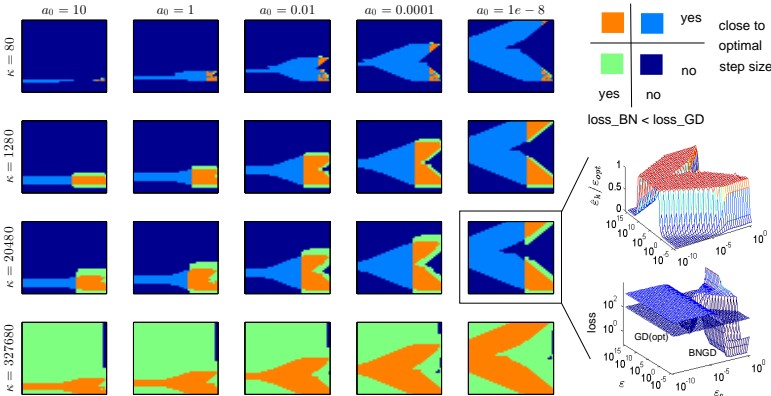

Figure 1: Compare of BNGD and GD on OLS model. The results are encoded by four different colors: whether $\hat{\varepsilon}_k$ is close to the optimal step size $\varepsilon_{opt}$ of GD, characterized by the inequality $0.8\varepsilon_{opt} < \hat{\varepsilon}_k < \varepsilon_{opt}/0.8$, and whether loss of BNGD is less than the optimal GD. Parameters: $H =$ diag(logspace(0,log10($\kappa$),100)), $u$ is randomly chosen uniformly from the unit sphere in $\mathbb{R}^{100}$, $w_0$ is set to $Hu/\|Hu\|$. The GD and BNGD iterations are executed for $k = 2000$ steps with the same $w_0$. In each image, the range of $\varepsilon_a$ (x-axis) is 1.99 * logspace(-10,0,41), and the range of $\varepsilon$ (y-axis) is logspace(-5,16,43).

## 4.2 EXPERIMENTS ON PRACTICAL DEEP LEARNING PROBLEMS

We conduct experiments on deep learning applied to standard classification datasets: MNIST (Le-Cun et al., 1998), Fashion MNIST (Xiao et al., 2017) and CIFAR-10 (Krizhevsky & Hinton, 2009). The goal is to explore if the key findings outlined at the beginning of this section continue to hold for more general settings. For the MNIST and Fashion MNIST dataset, we use two different networks: (1) a one-layer fully connected network (784 $\times$ 10) with softmax mean-square loss; (2) a four-layer convolution network (Conv-MaxPool-Conv-MaxPool-FC-FC) with ReLU activation function and cross-entropy loss. For the CIFAR-10 dataset, we use a five-layer convolution network (Conv-MaxPool-Conv-MaxPool-FC-FC-FC). All the trainable parameters are randomly initialized by the Glorot scheme (Glorot & Bengio, 2010) before training. For all three datasets, we use a minibatch size of 100 for computing stochastic gradients. In the BNGD experiments, batch normalization is performed on all layers, the BN parameters are initialized to transform the input to zero mean/unit variance distributions, and a small regularization parameter $\epsilon =$1e-3 is added to variance $\sqrt{\sigma^2 + \epsilon}$ to avoid division by zero.

**Scaling property** Theoretically, the scaling property 3.2 holds for any layer using BN. However, it may be slightly biased by the regularization parameter $\epsilon$. Here, we test the scaling property in practical settings. Figure 2 gives the loss of network-(2) (2CNN+2FC) at epoch=1 with different learning rate. The norm of all weights and biases are rescaled by a common factor $\eta$. We observe that the scaling property remains true for relatively large $\eta$. However, when $\eta$ is small, the norm of weights are small. Therefore, the effect of the $\epsilon$-regularization in $\sqrt{\sigma^2 + \epsilon}$ becomes significant, causing the curves to be shifted.

**Stability for large learning rates** We use the loss value at the end of the first epoch to characterize the performance of BNGD and GD methods. Although the training of models have generally not converged at this point, it is enough to extract some relative rate information. Figure 3 shows the loss value of the networks on the three datasets. It is observed that GD and BNGD with identical learning rates for weights and BN parameters exhibit a maximum allowed learning rate, beyond which the iterations becomes unstable. On the other hand, BNGD with separate learning rates exhibits a much larger range of stability over learning rate for non-BN parameters, consistent with our theoretical results in Theorem 3.3.

**Insensitivity of performance to learning rates** Observe that BN accelerates convergence more significantly for deep networks, whereas for one-layer networks, the best performance of BNGD and

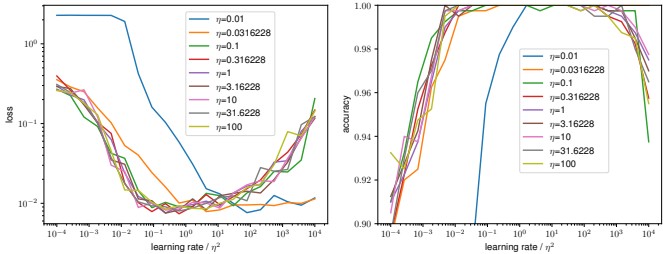

Figure 2: Tests of scaling property of the 2CNN+2FC network on MNIST dataset. BN is performed on all layers, and $\epsilon$=1e-3 is added to variance $\sqrt{\sigma^2 + \epsilon}$. All the trainable parameters (except the BN parameters) are randomly initialized by the Glorot scheme, and then multiplied by a same parameter $\eta$.

GD are similar. Furthermore, in most cases, the range of optimal learning rates in BNGD is quite large, which is in agreement with the OLS analysis (Proposition 3.5). This phenomenon is potentially crucial for understanding the acceleration of BNGD in deep neural networks. Heuristically, the "optimal" learning rates of GD in distinct layers (depending on some effective notion of "condition number") may be vastly different. Hence, GD with a shared learning rate across all layers may not achieve the best convergence rates for all layers at the same time. In this case, it is plausible that the acceleration of BNGD is a result of the decreased sensitivity of its convergence rate on the learning rate parameter over a large range of its choice.

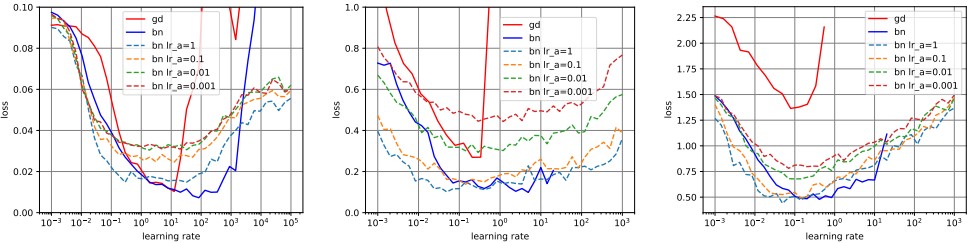

Figure 3: Performance of BNGD and GD method on MNIST (network-(1), 1FC), Fashion MNIST (network-(2), 2CNN+2FC) and CIFAR-10 (2CNN+3FC) datasets. The performance is characterized by the loss value at ephoch=1. In the BNGD method, both the shared learning rate schemes and separated learning rate scheme (learning rate lr_a for BN parameters) are given. The values are averaged over 5 independent runs.

## 5 CONCLUSION AND OUTLOOK

In this paper, we adopted a modeling approach to investigate the dynamical properties of batch normalization. The OLS problem is chosen as a point of reference, because of its simplicity and the availability of convergence results for gradient descent. Even in such a simple setting, we saw that BNGD exhibits interesting non-trivial behavior, including scaling laws, robust convergence properties, acceleration, as well as the insensitivity of performance to the choice of learning rates. Although these results are derived only for the OLS model, we show via experiments that these are qualitatively valid for general scenarios encountered in deep learning, and points to a concrete way in uncovering the reasons behind the effectiveness of batch normalization.

Interesting future directions include the extension of the results for the OLS model to more general settings of BNGD, where we believe the scaling law (Proposition 3.2) should play a significant role. In addition, we have not touched upon another empirically observed advantage of batch normalization, which is better generalization errors. It will be interesting to see how far the current approach takes us in investigating such probabilistic aspects of BNGD.

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

# A  PROOF OF THEOREMS

## A.1  GRADIENTS AND HESSIAN MATRIX

The objective function in problem (7) has an equivalent form:

$$J(a, w) = \tfrac{1}{2}(u - \tfrac{a}{\sigma}w)^T H(u - \tfrac{a}{\sigma}w) = \tfrac{1}{2}\|u\|_H^2 - \tfrac{w^T g}{\sigma}a + \tfrac{1}{2}a^2, \tag{16}$$

where $u = H^{-1}g$.

The gradients are:

$$\tfrac{\partial J}{\partial a} = -\tfrac{1}{\sigma}(w^T Hu - \tfrac{a}{\sigma}w^T Hw) = -\tfrac{1}{\sigma}w^T g + a, \tag{17}$$

$$\tfrac{\partial J}{\partial w} = -\tfrac{a}{\sigma}(Hu - \tfrac{a}{\sigma}Hw) + \tfrac{a}{\sigma^3}(w^T Hu - \tfrac{a}{\sigma}w^T Hw)Hw = -\tfrac{a}{\sigma}g + \tfrac{a}{\sigma^3}(w^T g)Hw. \tag{18}$$

The Hessian matrix is

$$\begin{pmatrix} \frac{\partial^2 J}{\partial g^2} & \frac{\partial^2 J}{\partial a \partial w} \\ \frac{\partial^2 J}{\partial w \partial a} & \frac{\partial^2 J}{\partial w^2} \end{pmatrix} = \begin{pmatrix} 1 & A_{21}^T \\ A_{21} & A_{22} \end{pmatrix} \tag{19}$$

where

$$A_{22} = \tfrac{a}{\sigma^3}(w^T g)\Big[H + \tfrac{1}{w^T g}\big((Hw)g^T + g(Hw)^T\big) - \tfrac{3}{\sigma^2}(Hw)(Hw)^T\Big], \tag{20}$$

$$A_{21} = -\tfrac{1}{\sigma}\big(g - \tfrac{1}{\sigma^2}(w^T g)Hw\big). \tag{21}$$

The objective function $J(a, w)$ has trivial critical points, $\{(a^*, w^*)|a^* = 0, w^{*T}g = 0\}$. It is obvious that $a^*$ is the minimizer of $J(a, w^*)$, but $(a^*, w^*)$ is not a local minimizer of $J(a, w)$ unless $g = 0$, hence $(a^*, w^*)$ are saddle points of $J(a, w)$. The Hessian matrix at those saddle points has at least a negative eigenvalue, i.e. the saddle points are strict. In fact, the eigenvalues at the saddle point $(a^*, w^*)$ are $\left\{\tfrac{1}{2}(1 \pm \sqrt{1 + 4\tfrac{\|g\|^2}{w^{*T}Hw^*}}), 0, ..., 0\right\}$ which contains $d-2$ repeated zero, a positive and a negative eigenvalue.

On the other hand, the nontrivial critical points satisfies the relations,

$$a^* = \pm\sqrt{u^T Hu}, w^* /\!/ u, \tag{22}$$

where the sign of $a^*$ depends on the direction of $u, w^*$, i.e. $sign(a^*) = sign(u^T w^*)$. It is easy to check that the nontrivial critical points are global minimizers. The Hessian matrix at those minimizers is $diag(1, H^*/\|w^*\|^2)$ where the matrix $H^*$ is

$$H^* = H - \frac{Huu^T H}{u^T Hu} \tag{23}$$

which is positive semi-definite and has a zero eigenvalue corresponding to the eigenvector $u$, i.e. $H^*u = 0$. The following lemma, similar to the well known Cauchy interlacing theorem, gives an estimate of eigenvalues of $H^*$.

**Lemma A.1.** *If $H$ is positive definite and $H^*$ is defined as $H^* = H - \frac{Huu^T H}{u^T Hu}$, then the eigenvalues of $H$ and $H^*$ satisfy the following inequalities:*

$$0 = \lambda_1(H^*) < \lambda_1(H) \le \lambda_2(H^*) \le \lambda_2(H) \le ... \le \lambda_d(H^*) \le \lambda_d(H). \tag{24}$$

*Here $\lambda_i(H)$ means the $i$-th smallest eigenvalue of $H$.*

*Proof.* (1) According to the definition, we have $H^*u = 0$, and for any $x \in \mathbb{R}^d$,

$$x^T H^* x = x^T Hx - \frac{(x^T Hu)^2}{u^T Hu} \in [0, x^T Hx], \tag{25}$$

which implies $H^*$ is semi-positive definite, and $\lambda_i(H^*) \ge \lambda_1(H^*) = 0$. Furthermore, we have the following equality:

$$x^T H^* x = \min_{t \in \mathbb{R}} \|x - tu\|_H^2. \tag{26}$$

(2) We will prove $\lambda_i(H^*) \leq \lambda_i(H)$ for all $i$, $1 \leq i \leq d$. In fact, using the Min-Max Theorem, we have

$$\lambda_i(H^*) = \min_{dimV=i} \max_{x \in V} \frac{x^T H^* x}{\|x\|^2} \leq \min_{dimV=i} \max_{x \in V} \frac{x^T H x}{\|x\|^2} = \lambda_i(H).$$

(3) We will prove $\lambda_i(H^*) \geq \lambda_{i-1}(H)$ for all $i$, $2 \leq i \leq d$. In fact, using the Max-Min Theorem, we have

$$\begin{aligned}
\lambda_i(H^*) &= \max_{dimV=n-i+1} \min_{x \in V} \frac{x^T H^* x}{\|x\|^2} = \max_{dimV=n-i+1, u \perp V} \min_{x \in V} \min_{t \in \mathbb{R}} \frac{\|x-tu\|_H^2}{\|x\|^2} \\
&\geq \max_{dimV=n-i+1, u \perp V} \min_{x \in V} \min_{t \in \mathbb{R}} \frac{\|x-tu\|_H^2}{\|x-tu\|^2} \\
&= \max_{dimV=n-i+1} \min_{y \in span\{V,u\}} \frac{\|y\|_H^2}{\|y\|^2}, y = x - tu \\
&\geq \max_{dimV=n-(i-1)+1} \min_{y \in V} \frac{y^T H y}{\|y\|^2} = \lambda_{i-1}(H),
\end{aligned}$$

where we have used the fact that $x \perp u$, $\|x - tu\|^2 = \|x\|^2 + t^2\|u\|^2 \geq \|x\|^2$. $\qquad\square$

There are several corollaries related to the spectral property of $H^*$. We first give some definitions. Since $H^*$ is positive semi-definite, we can define the $H^*$-seminorm.

**Definition A.2.** *The $H^*$-seminorm of a vector $x$ is defined as $\|x\|_{H^*} := x^T H^* x$. $\|x\|_{H^*} = 0$ if and only if $x$ is parallel to $u$.*

**Definition A.3.** *The pseudo-condition number of $H^*$ is defined as $\kappa^*(H^*) := \frac{\lambda_d(H^*)}{\lambda_2(H^*)}$.*

**Definition A.4.** *For any real number $\varepsilon$, the pseudo-spectral radius of the matrix $I - \varepsilon H^*$ is defined as $\rho^*(I - \varepsilon H^*) := \max_{2 \leq i \leq d} |1 - \varepsilon \lambda_i(H^*)|$.*

The following corollaries are direct consequences of Lemma A.1, hence we omit the proofs.

**Corollary A.5.** *The pseudo-condition number of $H^*$ is less than or equal to the condition number of $H$ :*

$$\kappa^*(H^*) := \frac{\lambda_d(H^*)}{\lambda_2(H^*)} \leq \frac{\lambda_d(H)}{\lambda_1(H)} =: \kappa(H), \tag{27}$$

*where the equality holds up if and only if $u \perp span\{v_1, v_d\}$, $v_i$ is the eigenvector of $H$ corresponding to eigenvalue $\lambda_i(H)$.*

**Corollary A.6.** *For any vector $x \in \mathbb{R}^d$ and any real number $\varepsilon$, we have $\|(I - \varepsilon H^*)x\|_{H^*} \leq \rho^*(I - \varepsilon H^*)\|x\|_{H^*}$.*

**Corollary A.7.** *For any positive number $\varepsilon > 0$, we have*

$$\rho^*(I - \varepsilon H^*) \leq \rho(I - \varepsilon H), \tag{28}$$

*where the inequality is strict if $u^T v_i \neq 0$ for $i = 1, d$.*

It is obvious that the inequality in (27) and (28) is strict for almost all $u$.

## A.2 SCALING PROPERTY

The dynamical system defined in equation (9)-(10) is completely determined by a set of configurations $\{H, u, a_0, w_0, \varepsilon_a, \varepsilon\}$. It is easy to check the system has the following scaling property:

**Lemma A.8** (Scaling property). *Suppose $\mu \neq 0, \gamma \neq 0, r \neq 0, Q^T Q = I$, then*

*(1) The configurations $\{\mu Q^T H Q, \frac{\gamma}{\sqrt{\mu}} Q u, \gamma a_0, \gamma Q w_0, \varepsilon_a, \varepsilon\}$ and $\{H, u, a_0, w_0, \varepsilon_a, \varepsilon\}$ are equivalent.*

*(2) The configurations $\{H, u, a_0, w_0, \varepsilon_a, \varepsilon\}$ and $\{H, u, a_0, rw_0, \varepsilon_a, r^2\varepsilon\}$ are equivalent.*

The scaling property is valid for general loss functions provided batch normalization is used. Consider a general problem

$$\min_{w \in \mathbb{R}^d} J_0(w) := E_{x,y}[f(y, x^T w)], \tag{29}$$

and its BN version

$$\min_{w \in \mathbb{R}^d, a \in \mathbb{R}} J(a, w) := E_{x,y}\big[f\big(y, aN_{BN}(x^T w)\big)\big]. \tag{30}$$

Then the gradient descent method gives the following iteration,

$$a_{k+1} = a_k + \varepsilon_a \frac{w_k^T \tilde{h}}{\sigma_k}, \tag{31}$$

$$w_{k+1} = w_k + \varepsilon \frac{a_k}{\sigma_k}\Big(\tilde{h} - \frac{w_k^T \tilde{h}}{\sigma_k^2} H w_k\Big), \tag{32}$$

where $\tilde{h} = h(a_k w_k / \sigma_k)$, and $h$ is the gradient of original problem:

$$h(w) := E_{x,y}[x f_2'(y, x^T w)]. \tag{33}$$

It is easy to check the general BNGD has the following property:

**Lemma A.9** (General scaling property). *Suppose $r \neq 0$, then the configurations $\{w_0, \varepsilon, *\}$ and $\{r w_0, r^2 \varepsilon, *\}$ are equivalent. Here the sign * means other parameters.*

### A.3 PROOF OF THEOREM 3.3

Recall the BNGD iterations

$$a_{k+1} = a_k + \varepsilon_a\Big(\frac{w_k^T g}{\sigma_k} - a_k\Big),$$

$$w_{k+1} = w_k + \varepsilon \frac{a_k}{\sigma_k}\Big(g - \frac{w_k^T g}{\sigma_k^2} H w_k\Big).$$

The scaling property simplify our analysis by allowing us to set, for example, $\|u\| = 1$ and $\|w_0\| = 1$. In the rest of this section, we only set $\|u\| = 1$.

For the step size of $a$, it is easy to check that $a_k$ tends to infinity with $\varepsilon_a > 2$ and initial value $a_0 = 1, w_0 = u$. Hence we only consider $0 < \varepsilon_a < 2$, which make the iteration of $a_k$ bounded by some constant $C_a$.

**Lemma A.10** (Boundedness of $a_k$). *If the step size $0 < \varepsilon_a < 2$, then the sequence $a_k$ is bounded for any $\varepsilon > 0$ and any initial value $(a_0, w_0)$.*

*Proof.* Define $\alpha_k := \frac{w_k^T g}{\sigma_k}$, which is bounded by $|\alpha_k| \leq \sqrt{u^T H u} =: C$, then

$$a_{k+1} = (1 - \varepsilon_a) a_k + \varepsilon_a \alpha_k$$
$$= (1 - \varepsilon_a)^{k+1} a_0 + (1 - \varepsilon_a)^k \varepsilon_a \alpha_0 + \dots + (1 - \varepsilon_a) \varepsilon_a \alpha_{k-1} + \varepsilon_a \alpha_k.$$

Since $|1 - \varepsilon_a| < 1$, we have $|a_{k+1}| \leq |a_0| + 2C \sum_{i=0}^k |1 - \varepsilon_a|^i \leq |a_0| + 2C \frac{1}{1 - |1 - \varepsilon_a|}$. $\qquad\square$

According to the iterations (34), we have

$$u - \frac{w_k^T g}{\sigma_k^2} w_{k+1} = \Big(I - \varepsilon \frac{a_k}{\sigma_k} \frac{w_k^T g}{\sigma_k^2} H\Big)\Big(u - \frac{w_k^T g}{\sigma_k^2} w_k\Big). \tag{34}$$

Define

$$e_k := u - \frac{w_k^T g}{\sigma_k^2} w_k, \tag{35}$$

$$q_k := u^T H u - \frac{(w_k^T g)^2}{\sigma_k^2} = \|e_k\|_H^2 \geq 0, \tag{36}$$

$$\hat{\varepsilon}_k := \varepsilon \frac{a_k}{\sigma_k} \frac{w_k^T g}{\sigma_k^2}, \tag{37}$$

and using the property $\frac{w^T g}{\sigma_k^2} = \underset{t}{\operatorname{argmin}} \|u - tw\|_H$, and the property of $H$-norm, we have

$$q_{k+1} \leq \left\| u - \frac{w_k^T g}{\sigma_k^2} w_{k+1} \right\|_H^2 = \|(I - \hat{\varepsilon}_k H)e_k\|_H^2 \leq \rho(I - \hat{\varepsilon}_k H)^2 q_k. \tag{38}$$

Therefore we have the following lemma to make sure the iteration converge:

**Lemma A.11.** *Let $0 < \varepsilon_a < 2$. If there are two positive numbers $\varepsilon^-$ and $\hat{\varepsilon}^+$, and the effective step size $\hat{\varepsilon}_k$ satisfies*

$$0 < \frac{\varepsilon^-}{\|w_k\|^2} \leq \hat{\varepsilon}_k \leq \hat{\varepsilon}^+ < \frac{2}{\lambda_{max}} \tag{39}$$

*for all $k$ large enough, then the iterations (34) converge to a minimizer.*

*Proof.* Without loss of generality, we assume $\frac{\varepsilon^-}{\|w_k\|^2} < \frac{1}{\lambda_{max}}$ and the inequality (39) is satisfied for all $k \geq 0$. We will prove $\|w_k\|$ converges and the direction of $w_k$ converges to the direction of $u$.

(1) Since $\|w_k\|$ is always increasing, we only need to prove it is bounded. We have,

$$\|w_{k+1}\|^2 = \|w_k\|^2 + \varepsilon^2 \frac{a_k^2}{\sigma_k^2} \|He_k\|^2 \tag{40}$$

$$= \|w_0\|^2 + \varepsilon^2 \sum_{i=0}^{k} \frac{a_i^2}{\sigma_i^2} \|He_i\|^2 \tag{41}$$

$$\leq \|w_0\|^2 + \varepsilon^2 \lambda_{max} \sum_{i=0}^{k} \frac{a_i^2}{\sigma_i^2} q_i \tag{42}$$

$$\leq \|w_0\|^2 + \varepsilon^2 \frac{\lambda_{max} C_a^2}{\lambda_{min}} \sum_{i=0}^{k} \frac{q_i}{\|w_i\|^2}. \tag{43}$$

The inequality in last lines are based on the fact that $\|He_i\|^2 \leq \lambda_{max} \|e_i\|_H^2$, and $|a_k|$ are bounded by a constant $C_a$. Next, we will prove $\sum_{i=0}^{\infty} \frac{q_i}{\|w_i\|^2} < \infty$, which implies $\|w_k\|$ are bounded.

According to the estimate (38), we have

$$q_{k+1} \leq \max_i \{|1 - \hat{\varepsilon}^+ \lambda_i|^2, |1 - \frac{\varepsilon^- \lambda_i}{\|w_k\|^2}|^2\} q_k \tag{44}$$

$$\leq \max\{1 - \gamma^+, 1 - \frac{\varepsilon^- \lambda_{min}}{\|w_k\|^2}\} q_k, \tag{45}$$

where $1 - \gamma^+ = \max_i \{|1 - \hat{\varepsilon}^+ \lambda_i|^2\} \in (0, 1)$. Using the definition of $q_k$, we have

$$q_k - q_{k+1} \geq \frac{\min\{\gamma^+ \|w_0\|^2, \varepsilon^- \lambda_{min}\}}{\|w_k\|^2} q_k =: \frac{Cq_k}{\|w_k\|^2} \geq 0. \tag{46}$$

Since $q_k$ is bounded in $[0, u^T Hu]$, summing both side of the inequality, we get the bound of the infinite series $\sum_k \frac{q_k}{\|w_k\|^2} \leq \frac{u^T Hu}{C} < \infty$.

(2) Since $\|w_k\|$ is bounded, we denote $\hat{\varepsilon}^- := \frac{\varepsilon^-}{\|w_\infty\|^2}$, and define $\rho := \max_i \{|1 - \hat{\varepsilon}^\pm \lambda_i|\} \in (0, 1)$, then the inequality (38) implies $q_{k+1} \leq \rho^2 q_k$. As a consequence, $q_k$ tends to zero, which implies the direction of $w_k$ converges to the direction of $u$.

(3) The convergence of $a_k$ is a consequence of $w_k$ converging.

$\square$

Since $a_k$ is bounded, we assume $|a_k| < \tilde{C}_a \sqrt{u^T Hu}$, $\tilde{C}_a \geq 1$, and define $\varepsilon_0 := \frac{1}{2\tilde{C}_a \kappa \lambda_{max}}$. The following lemma gives the convergence for small step size.

**Lemma A.12.** *If the initial values $(a_0, w_0)$ satisfies $a_0 w_0^T g > 0$, and step size satisfies $\varepsilon_a \in (0, 1], \varepsilon/\|w_0\|^2 < \varepsilon_0$, then the sequence $(a_k, w_k)$ converges to a global minimizer.*

**Remark 1:** If we set $a_0 = 0$, then we have $w_1 = w_0, a_1 = \varepsilon_a \frac{w_0^T g}{\sigma_0}$, hence $a_1 w_1^T g > 0$ provided $w_0^T g \neq 0$.

**Remark 2:** For the case of $\varepsilon_a \in (1, 2)$, if the initial value satisfies an additional condition $0 < |a_0| \leq \varepsilon_a \frac{|w_0^T g|}{\sigma_0}$, then we have $(a_k, w_k)$ converges to a global minimizer as well.

*Proof.* Without loss of generality, we only consider the case of $a_0 > 0, w_0^T g > 0, \|w_0\| \geq 1$.

(1) We will prove $a_k > 0, w_k^T g > 0$ for all $k$. Denote $y_k := w_k^T g, \delta = \frac{\|g\|}{4\kappa}$.

On the one hand, if $a_k > 0, 0 < y_k < 2\delta$, then

$$y_{k+1} \geq y_k + \varepsilon \frac{a_k}{\sigma_k} \frac{\|g\|^2}{2} \geq y_k. \tag{47}$$

On the other hand, when $a_k > 0, y_k > 0, \varepsilon < \varepsilon_0$, we have

$$y_{k+1} \geq \varepsilon \frac{a_k \|g\|^2}{\sigma_k} + y_k \left(1 - \varepsilon \frac{a_k}{\sigma_k^2} \sqrt{g^T H g}\right) \geq \tfrac{1}{2} y_k, \tag{48}$$

$$a_{k+1} \geq \min\{a_k, y_k/\sigma_k\}. \tag{49}$$

As a consequence, we have $a_k > 0, y_k \geq \delta_y := \min\{y_0, \delta\}$ for all $k$ by induction.

(2) We will prove the effective step size $\hat{\varepsilon}_k$ satisfying the condition in Lemma A.11.

Since $a_k$ is bounded, $\varepsilon < \varepsilon_0$, we have

$$\hat{\varepsilon}_k := \varepsilon \frac{a_k}{\sigma_k} \frac{w_k^T g}{\sigma_k^2} \leq \frac{\varepsilon \tilde{C}_a \lambda_{max}}{\lambda_{min} \|w_k\|^2} \leq \varepsilon \tilde{C}_a \kappa =: \hat{\varepsilon}^+ < \frac{1}{2\lambda_{max}}, \tag{50}$$

and

$$q_{k+1} \leq (1 - \hat{\varepsilon}_k \lambda_{min})^2 q_k \leq (1 - \hat{\varepsilon}_k \lambda_{min}) q_k < q_k. \tag{51}$$

which implies $\frac{w_{k+1}^T g}{\sigma_{k+1}} \geq \frac{w_k^T g}{\sigma_k} \geq \frac{w_0^T g}{\sigma_0}$. Furthermore, we have $a_k \geq \min\{a_0, \frac{w_0^T g}{\sigma_0}\}$, and there is a positive constant $\varepsilon^- > 0$ such that

$$\hat{\varepsilon}_k \geq \varepsilon \frac{a_k}{\lambda_{max} \|w_k\|^2} \frac{w_k^T g}{\sigma_k} \geq \frac{\varepsilon^-}{\|w_k\|^2}. \tag{52}$$

(3) Employing the Lemma A.11, we conclude that $(a_k, w_k)$ converges to a global minimizer. □

**Lemma A.13.** *If step size satisfies $\varepsilon_a \in (0, 1], \varepsilon/\|w_0\|^2 < \varepsilon_0$, then the sequence $(a_k, w_k)$ converges.*

*Proof.* Thanks to Lemma A.12, we only need to consider the case of $a_k w_k^T g \leq 0$ for all $k$, and we will prove the iteration converges to a saddle point in this case. Since the case of $a_k = 0$ or $w_k^T g = 0$ is trivial, we assume $a_k w_k^T g < 0$ below. More specifically, we will prove $|a_{k+1}| < r|a_k|$ for some constant $r \in (0, 1)$, which implies convergence to a saddle point.

(1) If $a_k$ and $a_{k+1}$ have same sign, hence different sign with $w_k^T g$, then we have $|a_{k+1}| = |1 - \varepsilon_a \| a_k| - \varepsilon_a |w_k^T g|/\sigma_k \leq |1 - \varepsilon_a\| a_k|$.

(2) If $a_k$ and $a_{k+1}$ have different signs, then we have

$$\frac{|w_k^T g|}{|a_k \sigma_k|} \leq \varepsilon \frac{1}{\sigma_k^2} \left(\|g\|^2 - \frac{w_k^T g}{\sigma_k^2} g^T H w_k\right) \leq 2\varepsilon \kappa \lambda_{max} < 1. \tag{53}$$

Consequently, we get

$$\frac{|a_{k+1}|}{|a_k|} = \varepsilon_a \frac{|w_k^T g|}{|a_k \sigma_k|} - (1 - \varepsilon_a) \leq 2\varepsilon \varepsilon_a \kappa \lambda_{max} - (1 - \varepsilon_a) < \varepsilon_a \leq 1. \tag{54}$$

(3) Setting $r := \max(|1 - \varepsilon_a|, 2\varepsilon\varepsilon_a \kappa \lambda_{max} - (1 - \varepsilon_a))$, we finish the proof. □

To simplify our proofs for Theorem 3.3, we give two lemmas which are obvious but useful.

**Lemma A.14.** *If positive series $f_k, h_k$ satisfy $f_{k+1} \leq r f_k + h_k, r \in (0, 1)$ and $\lim_{k \to \infty} h_k = 0$, then $\lim_{k \to \infty} f_k = 0$.*

*Proof.* It is obvious, because the series $b_k$ defined by $b_{k+1} = rb_k + h_k, b_0 > 0$, tends to zeros. $\quad\square$

**Lemma A.15** (Separation property). *For $\delta_0$ small enough, the set $S := \{w | y^2 q < \delta_0, \|w\| \geq 1\}$ is composed by two separated parts: $S_1$ and $S_2$, $dist(S_1, S_2) > 0$, where in the set $S_1$ one has $y^2 < \delta_1, q > \delta_2$, and in $S_2$ one has $q < \delta_2, y^2 > \delta_1$ for some $\delta_1 > 0, \delta_2 > 0$. Here $y := w^T g, q := u^T H u - \frac{(w^T H u)^2}{w^T H w} = u^T H u - \frac{y^2}{w^T H w}$.*

*Proof.* The proof is based on $H$ being positive. The geometric meaning is illustrated in Figure 4. $\quad\square$

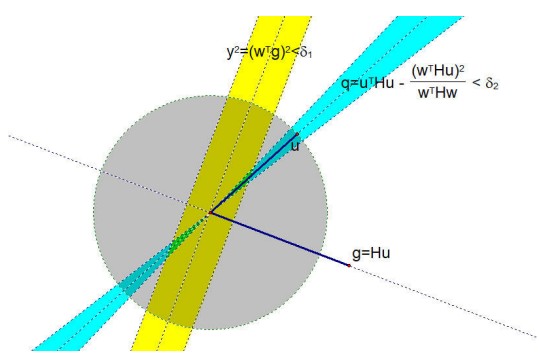

Figure 4: The geometric meaning of the separation property

**Corollary A.16.** *If $\lim_{k \to \infty} \|w_{k+1} - w_k\| = 0$, and $\lim_{k \to \infty} (w_k^T g)^2 q_k = 0$, then either $\lim_{k \to \infty} (w_k^T g)^2 = 0$ or $\lim_{k \to \infty} q_k = 0$.*

*Proof.* Denote $y_k := w_k^T g$. According to the separation property (Lemma A.15), we can chose a $\delta_0 > 0$ small enough such that the separated parts of the set $S := \{w | y^2 q < \delta_0, \|w\| \geq 1\}$, $S_1$ and $S_2$, have $dist(S_1, S_2) > 0$.

Because $y_k^2 q_k$ tends to zero, we have $w_k$ belongs to $S$ for $k$ large enough, for instance $k > k_1$. On the other hand, because $\|w_{k+1} - w_k\|$ tends to zero, we have $\|w_{k+1} - w_k\| < dist(S_1, S_2)$ for $k$ large enough, for instance $k > k_2$. Then consider $k > k_3 := \max(k_1, k_2)$, we have all $w_k$ belongs to the same part $S_1$ or $S_2$.

If $w_k \in S_1, (q_k > \delta_2)$, for all $k > k_3$, then we have $\lim_{k \to \infty} (w_k^T g)^2 = 0$.

On the other hand, if $w_k \in S_2, (y_k^2 > \delta_1)$, for all $k > k_3$, then we have $\lim_{k \to \infty} q_k = 0$.

$\quad\square$

**Theorem A.17.** *Let $\varepsilon_a \in (0, 1]$ and $\varepsilon > 0$. The sequence $(a_k, w_k)$ converges for any initial value $(a_0, w_0)$.*

*Proof.* We will prove $\|w_k\|$ converges, then prove $(a_k, w_k)$ converges as well.

(1) We will prove that $\|w_k\|$ is bounded and hence converges.

In fact, according to the Lemma A.13, once $\|w_k\|^2 \geq \varepsilon/\varepsilon_0$ for some $k$, the rest of the iteration will converge, hence $\|w_k\|$ is bounded.

(2) We will prove $\lim_{k \to \infty} \|w_{k+1} - w_k\| = 0$, and $\lim_{k \to \infty} (w_k^T g)^2 q_k = 0$.

The convergence of $\|w_k\|$ implies $\sum_k a_k^2 q_k$ is summable. As a consequence,

$$\lim_{k \to \infty} a_k^2 p_k = 0, \ \lim_{k \to \infty} a_k e_k = 0, \tag{55}$$

and $\lim_{k \to \infty} \|w_{k+1} - w_k\| = 0$. In fact, we have

$$\|w_{k+1} - w_k\|^2 = \varepsilon^2 \frac{a_k^2}{\sigma^2} \|He_k\|^2 \leq \frac{\lambda_{max} \varepsilon^2}{\lambda_{min}^2} a_k^2 q_k \to 0. \tag{56}$$

Consider the iteration of series $|a_k - w_k^T g / \sigma_k|$,

$$
\begin{aligned}
\left| a_{k+1} - \frac{w_{k+1}^T g}{\sigma_{k+1}} \right| &\leq \left| a_{k+1} - \frac{w_{k+1}^T g}{\sigma_k} \right| + \left| \frac{w_{k+1}^T g}{\sigma_k} - \frac{w_{k+1}^T g}{\sigma_{k+1}} \right| \\
&\leq (1 - \varepsilon_a) \left| a_k - \frac{w_k^T g}{\sigma_k} \right| + \varepsilon \frac{|a_k g^T He_k|}{\sigma_k^2} + \frac{|w_{k+1}^T g|}{(\sigma_k \sigma_{k+1})} |\sigma_{k+1} - \sigma_k| \\
&\leq (1 - \varepsilon_a) \left| a_k - \frac{w_k^T g}{\sigma_k} \right| + \varepsilon \frac{\|g\|_H \|a_k e_k\|_H}{\sigma_k^2} + \frac{|w_{k+1}^T g|}{(\sigma_k \sigma_{k+1})} \varepsilon \frac{\lambda_{max}}{\sigma_k} \|a_k e_k\|_H \\
&\leq (1 - \varepsilon_a) \left| a_k - \frac{w_k^T g}{\sigma_k} \right| + 2C \|a_k e_k\|_H. 
\end{aligned}
\tag{57}
$$

The constant $C$ in (57) can be chosen as $C = \frac{\varepsilon \lambda_{max} \|u\|_H}{\lambda_{min} \|w_0\|^2}$. Since $\|a_k e_k\|_H$ tends to zero, we can use Lemma A.14 to get $\lim_{k \to \infty} |a_k - w_k^T g / \sigma_k| = 0$. Combine the equation (55), then we have $\lim_{k \to \infty} (w_k^T g)^2 p_k = 0$.

(3) According to the Corollary A.16, we have either $\lim_{k \to \infty} y_k^2 = 0$, or $\lim_{k \to \infty} q_k = 0$. In the former case, the iteration of $(a_k, w_k)$ converges to a saddle point. However, in the latter case, $(a_k, w_k)$ converges to a global minimizer. In both cases we have $(a_k, w_k)$ converges.

$\square$

To finish the proof of Theorem 3.3, we have to demonstrate the special case of $\varepsilon_a = 1$ where the set of initial values such that BN iteration converges to saddle points is Lebeguse measure zero. We leave this demonstration in next section where we consider the case of $\varepsilon_a \geq 1$.

### A.4 Impossibility of converging to strict saddle points

In this section, we will prove the set of initial values such that BN iteration converges to saddle points is (Lebeguse) measure zero, as long as $\varepsilon_a \geq 1$. The tools in our proof is similar to the analysis of gradient descent on non-convex objectives (Lee et al., 2016; Panageas & Piliouras, 2017). In addition, we used the real analytic property of the BN loss function (16).

For brevity, here we denote $x := (a, w)$ and let $\varepsilon_a = \varepsilon$, then the BN iteration can be rewrote as

$$x_{n+1} = T(x_n) := x_n - \varepsilon \nabla J(x_n).$$

**Lemma A.18.** *If $A \subset T(\mathbb{R}^d / \{0\})$ is a measure zero set, then the preimage $T^{-1}(A)$ is of measure zero as well.*

*Proof.* Since $T$ is smooth enough, according to Theorem 3 of (Ponomarev, 1987), we only need to prove the Jacobian of $T(x)$ is nonzero for almost all $x \in \mathbb{R}^d$. In other words, the set $\{x : \det(I - \varepsilon \nabla^2 J(x)) = 0\}$ is of measure zero. This is true because the function $\det(I - \varepsilon \nabla^2 J(x))$ is a real analytic function of $x \in \mathbb{R}^d / \{0\}$. (Details of properties of real analytic functions can be found in (Krantz & Parks, 2002) for instance).

$\square$

**Lemma A.19.** *Let $f : X \to \mathbb{R}$ be twice continuously differentiable in an open set $X \subset \mathbb{R}^d$ and $x^* \in X$ be a stationary point of $f$. If $\varepsilon > 0$, $\det(I - \varepsilon \nabla^2 f(x^*)) \neq 0$ and the matrix $\nabla^2 f(x^*)$ has at least a negative eigenvalue, then there exist a neighborhood $U$ of $x^*$ such that the following set $B$ has measure zero,*

$$B := \{x_0 \in U : x_{n+1} = x_n - \varepsilon \nabla f(x_n) \in U, \forall n \geq 0\}. \tag{58}$$

*Proof.* The detailed proof is similar to (Lee et al., 2016; Panageas & Piliouras, 2017).

Define the transform function as $F(x) := x - \varepsilon \nabla f(x)$. Since $\det(I - \varepsilon \nabla^2 f(x^*)) \neq 0$, accorded to the inverse function theorem, there exist a neighborhood $U$ of $x^*$ such that $T$ has differentiable inverse. Hence $T$ is a local $C^1$ diffeomorphism, which allow us to use the central-stable manifold theorem (Shub, 2013). The negative eigenvalues of $\nabla^2 f(x^*)$ indicates $\lambda_{max}(I - \varepsilon \nabla^2 f(x^*)) > 1$ and the dimension of the unstable manifold is at least one, which implies the set $B$ is on a lower dimension manifold hence $B$ is of measure zero.

$\square$

**Lemma A.20.** *If $\varepsilon_a = \varepsilon \geq 1$, then the set of initial values such that BN iteration converges to saddle points is of Lebeguse measure zero.*

*Proof.* We will prove this argument using Lemma A.18 and Lemma A.19. Denote the saddle points set as $W := \{(a^*, w^*) : a^* = 0, w^{*T} g = 0\}$. The basic point is that the saddle point $x^* := (a^*, w^*)$ of the BN loss function (16) has eigenvalues $\left\{ \frac{1}{2}(1 \pm \sqrt{1 + 4\frac{\|g\|^2}{w^{*T} H w^*}}), 0, ..., 0 \right\}$ of the Hessian matrix.

(1) For each saddle point $x^* := (a^*, w^*)$ of BN loss function, $\varepsilon \geq 1$ is enough to allow us to use Lemma A.19. Hence there exist a neighborhood $U_{x^*}$ of $x^*$ such that the following set $B_{x^*}$ is of measure zero,

$$B_{x^*} := \{x_0 \in U_{x^*} : x_n \in U_{x^*}, \forall n \geq 1\}. \tag{59}$$

(2) The neighborhoods $U_{x^*}$ of all $x^* \in W$ forms a cover of $W$, hence, accorded to Lindelöf's open cover lemma, there are countable neighborhoods $\{U_i : i = 1, 2, ...\}$ cover $W$, i.e. $U := \cup_i U_i \supseteq W$. As a consequence, the following set $A_0$ is of measure zero,

$$A_0 := \cup_i B_i = \cup_i \{x_0 \in U_i : x_n \in U_i, \forall n \geq 1\}. \tag{60}$$

(3) Define $A_{m+1} := T^{-1}(A_m) = \{x \in \mathbb{R}^d : T(x) \in A_m\}, m \geq 0$. According to Lemma A.18, we have all $A_m$ and $\cup_m A_m$ are of measure zero.

(4) Since each initial value $x_0$ such that the iteration converges to a saddle point must be contained in some set $A_m$, we finish the proof.

$\square$

Combine the results of Lemma A.20, scaling property 3.2 and the convergence theorem A.17, we have the following theorem directly.

**Theorem A.21.** *If $\varepsilon_a = 1, \varepsilon \geq 0$, then the BN iteration (9)-(10) converges to global minimizers for almost all initial values.*

## A.5 CONVERGENCE RATE

In the last section, we encountered the following estimate for $e_k = u - \frac{w_k^T g}{\sigma_k^2} w_k$

$$\|e_{k+1}\|_H \leq \rho(I - \hat{\varepsilon}_k H)\|e_k\|_H. \tag{61}$$

We can improve the convergence rate of the above if $H^*$ has better spectral property. This is the content of Proposition 3.4 and the following lemma is enough to prove it.

**Lemma A.22.** *The following inequality holds,*

$$(1 - \delta_k)\|e_{k+1}\|_H \leq \left( \rho^*(I - \hat{\varepsilon}_k H^*) + \delta_k \right) \|e_k\|_H, \tag{62}$$

*where $\delta_k := \frac{\lambda_{max} \varepsilon |a_k|}{\sigma_k^2} \|e_k\|_H$.*

*Proof.* The case of $w_k^T g = 0$ is trivial, hence we assume $w_k^T g \neq 0$ in the following proof. Rewrite the iteration on $w_k$ as the following equality,

$$u - \frac{w_k^T g}{\sigma_k^2} w_{k+1} = (I - \hat{\varepsilon}_k H) e_k = (I - \hat{\varepsilon}_k H^*) e_k - \hat{\varepsilon}_k \left(1 - \frac{(w_k^T g)^2}{u^T H u \sigma_k^2}\right) H u. \tag{63}$$

Then we will use the properties of $H^*$-seminorm to prove our argument.

(1) Estimate the $H^*$-seminorm on the right hand of equation (63).

$$\|\text{right}\|_{H^*} \leq \|(I - \hat{\varepsilon}_k H^*) e_k\|_{H^*} + |\hat{\varepsilon}_k| \left(1 - \frac{(w_k^T g)^2}{u^T H u \sigma_k^2}\right) \|H u\|_{H^*} \tag{64}$$

$$\leq \rho^*(I - \hat{\varepsilon}_k H^*) \|e_k\|_{H^*} + \frac{\lambda_{max} |\hat{\varepsilon}_k|}{\sqrt{u^T H u}} \|e_k\|_H^2 \tag{65}$$

$$= \rho^*(I - \hat{\varepsilon}_k H^*) \frac{|w_k^T g|}{\sqrt{u^T H u} \sigma_k} \|e_k\|_H + \frac{\lambda_{max} \varepsilon |a_k w_k^T g|}{\sqrt{u^T H u} \sigma_k^3} \|e_k\|_H^2 \tag{66}$$

$$= \frac{|w_k^T g|}{\sqrt{u^T H u} \sigma_k} \left(\rho^*(I - \hat{\varepsilon}_k H^*) + \delta_k\right) \|e_k\|_H. \tag{67}$$

(2) Estimate the $H^*$-seminorm on the left hand of equation (63). Using the $H$-norm on the iteration of $w_k$, we have

$$\sigma_{k+1} = \|w_k + \varepsilon \frac{a_k}{\sigma_k} H e_k\|_H \geq \sigma_k - \varepsilon \frac{\lambda_{max} |a_k|}{\sigma_k} \|e_k\|_H. \tag{68}$$

Consequently, we have

$$\|\text{left}\|_{H^*} = \frac{|w_k^T g|}{\sqrt{u^T H u} \sigma_k} \frac{\sigma_{k+1}}{\sigma_k} \|e_{k+1}\|_H \geq \frac{|w_k^T g|}{\sqrt{u^T H u} \sigma_k} (1 - \delta_k) \|e_{k+1}\|_H. \tag{69}$$

(3) Combining (1) and (2), we finish the proof. $\qquad \square$

Now, we turn to the convergence of the loss function which can be rewritten as $J_k = \frac{1}{2} \|\tilde{e}_k\|_H^2$ with $\tilde{e}_k = u - \frac{a_k}{\sigma_k} w_k$. There is an useful equality between $\|\tilde{e}_k\|_H^2$ and $\|e_k\|_H^2$:

$$\|\tilde{e}_k\|_H^2 = \|e_k\|_H^2 + \left(a_k - \frac{w_k^T g}{\sigma_k}\right)^2. \tag{70}$$

Recalling the inequality (57) and the boundedness of $a_k$, we have a constant $C_0$ such that

$$\left|a_{k+1} - \frac{w_{k+1}^T g}{\sigma_{k+1}}\right| \leq |1 - \varepsilon_a| \left|a_k - \frac{w_k^T g}{\sigma_k}\right| + C_0 \|e_k\|_H, \tag{71}$$

which indicates that we can use the convergence of $e_k$ to estimate the convergence of the loss value $J_k$. In fact we have the following lemma.

**Lemma A.23.** *If $\|e_k\|_H \leq C \rho^k$ for some constant $C$ and $\rho \in (0, 1)$, $\varepsilon_a \in (0, 1]$, then we have*

$$\|\tilde{e}_k\|_H^2 \leq C^2 \rho^{2k} + \left(C_1 (1 - \varepsilon_a)^k + C_2 k \gamma^k\right)^2, \tag{72}$$

*where $\gamma = \max(\rho, 1 - \varepsilon_a)$, $C_1 = |a_0 - w_0^T g / \sigma_0|$ and $C_2 = C C_0$.*

*Proof.* According to the inequality (71), we have

$$\left|a_k - \frac{w_k^T g}{\sigma_k}\right| \leq C_1 (1 - \varepsilon_a)^k + C_2 \sum_{i=0}^{k-1} (1 - \varepsilon_a)^i \rho^{k-i} \leq C_1 (1 - \varepsilon_a)^k + C_2 k \gamma^k. \tag{73}$$

Put it in the equality (70), then we finish the proof.

$\qquad \square$

## A.6 ESTIMATING THE EFFECTIVE STEP SIZE

Accorded to Lemma A.12, the effective step size $\hat{\varepsilon}_k$ has same order with $\frac{\varepsilon}{\|w_k\|^2}$ provided $a_0 w_0^T g > 0, \varepsilon/\|w_0\| < \varepsilon_0$. In fact, we have

$$\frac{C_1 \varepsilon}{\|w_k\|^2} := \frac{a_0 w_0^T g}{\sigma_0} \frac{\varepsilon}{\lambda_{max} \|w_k\|^2} \leq \hat{\varepsilon}_k \leq \sqrt{u^T H u} \frac{C_a \varepsilon}{\lambda_{min} \|w_k\|^2} =: \frac{C_2 \varepsilon}{\|w_k\|^2}. \tag{74}$$

Hence, to prove the Proposition 3.5, we only need to estimate the norm of $w_k$.

*proof of Proposition 3.5.* According to the BNGD iteration, we have (see the proof of Lemma A.11)

$$\|w_{k+1}\|^2 \leq \|w_0\|^2 + \varepsilon^2 \lambda_{max} \sum_{i=0}^{k} \frac{a_i^2}{\sigma_i^2} q_i. \tag{75}$$

(1) When $\frac{\varepsilon}{\|w_0\|^2} < \varepsilon_0$ ($\varepsilon_0$ is defined in Lemma A.12), the sequence $q_k$ satisfies $q_{k+1} \leq (1 - \hat{\varepsilon}_k \lambda_{min}) q_k$. Hence the norm of $w_k$ is bounded by

$$\|w_k\|^2 \leq \|w_0\|^2 + \varepsilon \kappa C_a \frac{\sigma_0}{w_0^T g} \sum_{i=0}^{\infty} (q_i - q_{i+1}) \leq \|w_0\|^2 + C\varepsilon, \tag{76}$$

for some constant $C$. As a consequence,

$$\tilde{C}_1 \varepsilon := \frac{C_1 \varepsilon}{\|w_0\|^2 (1 + C\varepsilon_0)} \leq \hat{\varepsilon}_k \leq \frac{C_2 \varepsilon}{\|w_0\|^2} =: \tilde{C}_2 \varepsilon. \tag{77}$$

(2) When $\varepsilon$ is large enough, the increment of the norm $\|w_k\|$ at the first step is large as well. In fact, we have

$$\|w_1\|^2 - \|w_0\|^2 = \varepsilon^2 \frac{a_0^2}{\sigma_0^2} \|He_0\|^2 = C_3 \varepsilon^2. \tag{78}$$

Since $\|g\|^2 \geq \frac{w_0^T g}{\sigma_0^2} g^T H w_0$, we have $a_1 w_1^T g > a_1 w_0^T g > 0$. Choose $\varepsilon$ to be larger than some value $\varepsilon_1$ such that $\frac{\varepsilon}{\|w_1\|^2} < \varepsilon_0$, then we can use the argument in (1) on $(a_1, w_1)$. More precisely, there are two constants, $C_1, C_2$, such that

$$\frac{C_1 \varepsilon}{\|w_1\|^2} \leq \hat{\varepsilon}_k \leq \frac{C_2 \varepsilon}{\|w_1\|^2}. \tag{79}$$

Plugging the equation (78) into it, we have

$$\frac{C_1 \varepsilon_1^2}{\|w_0\|^2 + C_3 \varepsilon_1^2} \leq \frac{C_1 \varepsilon^2}{\|w_0\|^2 + C_3 \varepsilon^2} \leq \hat{\varepsilon}_k \varepsilon \leq \frac{C_2 \varepsilon^2}{\|w_0\|^2 + C_3 \varepsilon^2} \leq \frac{C_2}{C_3}. \tag{80}$$

$\square$

# B   MODIFIED BNGD

Through our analysis, we discovered that a modification of the BNGD, which we call MBNGD, becomes much easier to analyze and possesses better convergence properties. Note that the results in the main paper do not depend on the results in this section.

The modification is simply to enforce $a_k = \frac{w_k^T g}{\sigma_k}$ at every iteration, which yields the modified iterations:

$$w_{k+1} = w_k + \varepsilon \frac{w_k^T g}{\sigma_k^2}\Big(g - \frac{w_k^T g}{\sigma_k^2} H w_k\Big). \tag{81}$$

In a sense, one can view the above as a limiting version of BNGD where the BN rescaling variable $a$ is adjusted every step to the optimal value based on the current value of the weights $w$. The MBNGD iterations is governed by the variables: $H, u, a_0, w_0, \varepsilon$, where the scaling properties (Lemma 3.2, omit the parameter $\varepsilon_a$ now) remains. More importantly, we find the iteration will converge to a saddle point if and only if it exactly meets the saddle point at a finite step. More precisely, we have the convergence theorem:

**Theorem B.1** (Convergence for modified BNGD). *The iteration sequence $w_k$ in equation (81) converges for any initial value $w_0$ and any step size $\varepsilon > 0$. It converges to a global minimizer almost sure, in the sense that the set of initial values such that $w_k$ converges to a saddle point is of Lebesgue measure zero. Furthermore, It converges to a saddle point if and only if $w_k^T g = 0$ for some $k$.*

*Particularly, if $\varepsilon < \frac{2\|w_0\|^2}{\lambda_{max}\kappa\|u\|^2}, w_0^T g \neq 0$, then $w_k$ converges to a global minimizer.*

In the following, we assume $\|u\| = 1$.

## B.1   PROOF

**Lemma B.2.** *If $w_0^T g \neq 0$ and $\frac{\varepsilon}{\|w_0\|^2} < \varepsilon_0 := \frac{2}{\kappa\lambda_{max}}$, then the sequence $w_k$ converges to a global minimizer.*

*Proof.* Similar to the proof of Lemma A.12, but here the effective step size is always nonnegative which is defined as

$$\hat{\varepsilon}_k := \varepsilon\Big(\frac{w_k^T g}{\sigma_k^2}\Big)^2 \leq \frac{\varepsilon\kappa}{\|w_k\|^2} \leq \frac{\varepsilon\kappa}{\|w_0\|^2} =: \hat{\varepsilon}^+ < \frac{2}{\lambda_{max}}. \tag{82}$$

The inequality (38) immediately gives $q_{k+1} \leq q_k$, which implies $\frac{(w_{k+1}^T g)^2}{\sigma_{k+1}^2} \geq \frac{(w_k^T g)^2}{\sigma_k^2} \geq \frac{(w_0^T g)^2}{\sigma_0^2}$. As a consequence, the effective step size has a lower bound

$$\hat{\varepsilon}_k \geq \varepsilon\frac{(w_0^T g)^2}{\sigma_0^2}\frac{1}{\lambda_{max}\|w_k\|^2} =: \frac{\varepsilon^-}{\|w_k\|^2}. \tag{83}$$

Employing the Lemma A.11, we conclude that $w_k$ converges to a global minimizer. □

**Lemma B.3.** *If $w_k^T g \neq 0$ for all $k$, then $\sum_{k=0}^{\infty} |w_k^T g| = \sum_{k=0}^{\infty} (w_k^T g)^2 = \infty$.*

*Proof.* Without loss of generality, we assume $\|w_0\| \geq 1$, denote $y_k := w_k^T g$ and set $\delta = \frac{\|g\|}{4\kappa}$. From the iteration of $w_k$, we have

$$y_{k+1} = y_k + \epsilon\frac{y_k}{\sigma_k^2}\Big(\|g\|^2 - \frac{y_k}{\sigma_k^2}g^T H w_k\Big). \tag{84}$$

If $0 < |w_k^T g| < 2\delta$, then we have the inequality:

$$\|g\|^2 - \frac{y_k}{\sigma_k^2}g^T H w_k \geq \|g\|^2 - \frac{\kappa\|g\|}{\|w_k\|}|y_k| \geq \frac{1}{2}\|g\|^2, \tag{85}$$

then

$$|y_{k+1}| \geq \Big(1 + \frac{\epsilon}{\lambda_{max}\|w_k\|^2}\frac{\|g\|^2}{2}\Big)|y_k| > |y_k| > 0. \tag{86}$$

As a consequence, $\lim_{k\to\infty} w_k^T g = 0$ is not possible unless $w_k^T g = 0$ for some $k$, which implies the results we want.

□

□

**Theorem B.4.** *The iteration sequence $w_k$ in equation (81) converges for any initial value $w_0$ and any step size $\varepsilon > 0$. Furthermore, $w_k$ will converge to a global minimizer unless $w_k^T g = 0$ for some $k$.*

*Proof.* Obviously, if $w_k^T g = 0$ for some $k = k_0$, then $w_k = w_{k_0}$ for all $k \geq k_0$, hence $w_k$ converges to $w_{k_0}$. Without losing generality, we consider $w_k^T g \neq 0$ for all $k$ and $\|w_0\| \geq 1$ below.

(1) Firstly, we will prove that $\|w_k\|$ is bounded and hence converges.

In fact, according to the Lemma B.2, once $\|w_k\|^2 \geq \varepsilon/\varepsilon_0$ for some $k$, the rest of the iteration will converge, hence $\|w_k\|$ is bounded.

(2) Secondly, we will prove $w_k$ converges to a vector parallel to $u$.

Denote $y_k := w_k^T g$, $z_k := \frac{w_k^T g}{\sigma^2}$. The convergence of $\|w_k\|$ indicates that $\sum_{k=0}^{\infty} z_k^2 q_k$ is summable, and then $\sum_{k=0}^{\infty} y_k^2 q_k$ is summable as well. Therefore we have

$$\|w_{k+1} - w_k\|^2 = \varepsilon^2 \frac{(w_k^T g)^2}{\sigma^4} \left\| g - \frac{w_k^T g}{\sigma_k^2} H w_k \right\|^2 \leq \lambda_{max} \varepsilon^2 z_k^2 q_k, \tag{87}$$

and the above tends to zero, i.e. $\lim_{k \to \infty} \|w_{k+1} - w_k\| = 0$.

According to the separation property (Lemma A.15), we can chose a $\delta_0 > 0$ small enough such that the separated parts of the set $S := \{w | y^2 q < \delta_0, \|w\| \geq 1\}$, $S_1$ and $S_2$, have $dist(S_1, S_2) > 0$.

Because $y_k^2 q_k$ tends to zero, we have $w_k$ belongs to $S$ for $k$ large enough, for instance $k > k_1$. On the other hand, because $\|w_{k+1} - w_k\|$ tends to zero, we have $\|w_{k+1} - w_k\| < dist(S_1, S_2)$ for $k$ large enough, for instance $k > k_2$. Then consider $k > k_3 := \max(k_1, k_2)$, we have all $w_k$ belongs to the same part $S_1$ or $S_2$.

However, Lemma B.3 says $\sum_{k=0}^{\infty} y_k^2 = \infty$, hence $w_k \in S_1$ ($q_k > \delta_2$) for all $k > k_3$ is not true. Therefore $w_k \in S_2$ ($y_k^2 > \delta_1$) for all $k > k_3$. Consequently, we can claim that $\sum_{k=0}^{\infty} q_k$ is summable and $w_k$ converges to a vector parallel to $u$.

$\square$

## B.2 EXPERIMENT

Here we test the convergence and stability of MBNGD for OLS model. Consider the diagonal matrix $H = diag(h)$, where $h = (1, ..., \kappa)$ is an increasing sequence. The scaling property allows us to set the initial value $w_0$ having same 2-norm with $u$, $\|w_0\| = \|u\| = 1$.

Figure 5 gives an example of a 5-dimensional $H$ with condition number $\kappa = 2000$. The GD and MBNGD iteration are executed $k = 5000$ times where $u$ and $w_0$ are randomly chosen from the unit sphere. The values of effective step size, loss $\|e_k\|_H^2$ and error $\|e_k\|$ are plotted. Furthermore, to explore the performance of GD and MBNGD, the mean values over 300 random tests are given. It is worth to note that, the geometric mean (G-mean) is more reliable than the arithmetic mean (A-mean), where the geometric mean of $x$ can be defined as $\exp(\mathbb{E}(\ln x))$. Here the reliability means that the G-mean converges quickly when the number of tests increase, however the A-mean does not converge as quickly. In this example, the optimal convergence rate of MBNGD is observably better than GD. This acceleration phenomenon is ascribed to the pseudo-condition number of $\kappa^*(H^*)$ being less than $\kappa(H)$. However, if the difference between (pseudo-)condition number of $H$ and $H^*$ is small, the acceleration is imperceptible.

Another important observation is that the BN significantly extends the range of 'optimal' step size, which is embodied by the effective step size $\hat{\varepsilon}_k$ having a large constant $C$ in $\hat{\varepsilon} = O(C\varepsilon^{-1})$. This means we can chose step size in BN at a large interval to get almost same or better convergence rate than that of the best choice for GD.

Figure 6 gives an example of 100-dimension $H$ with condition number $\kappa = 2000$. Similar results as those in the 5-dimensional case are obtained. However, the best optimal convergence rate of MBNGD here has not noticeably improved compared with GD with the optimal learning rate, which is due to the fact that large $d$ decrease the difference between eigenvalues of $H$ and $H^*$.

Additional tests indicate that:

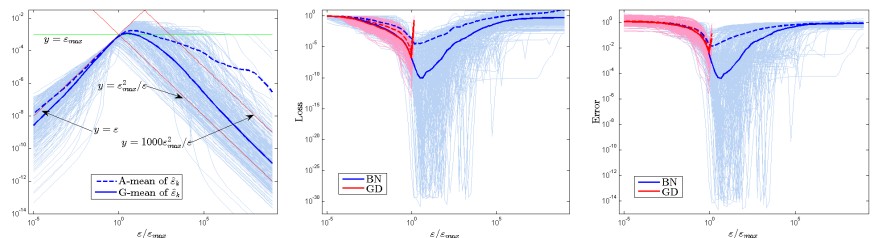

Figure 5: Plot of 300 random initial tests. H = diag(logspace(0,log10(2000),5)).

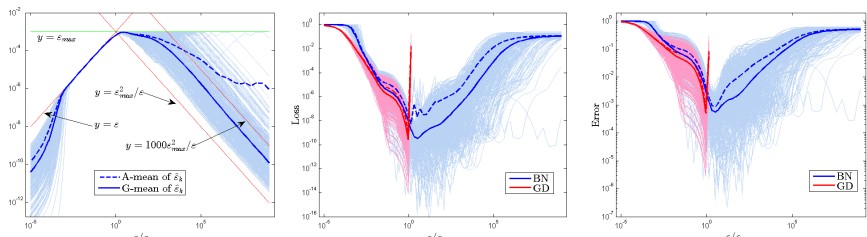

Figure 6: Plot of 500 random initial tests. H = diag(linspace(1,2000,100)).

(1) larger dimensions leads to larger intervals of 'optimal' step size, (Figure 7)

(2) the effect of condition number on the 'optimal' interval is small (Figure 8).

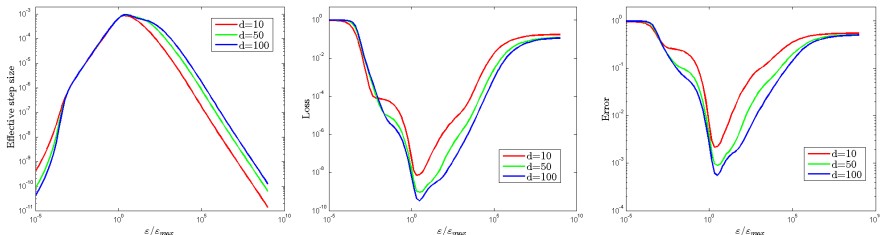

Figure 7: H = diag(linspace(1,2000,d)).

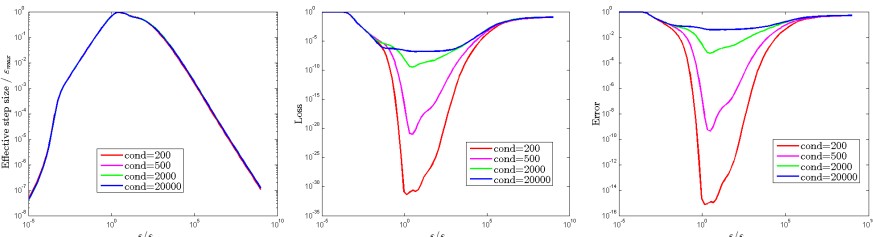

Figure 8: H = diag(linspace(1,cond,100)).

