# OpenReview forum: "On the Convergence and Robustness of Batch Normalization"
_ICLR.cc/2019/Conference_

### Official Review · AnonReviewer1 · 2018-11-02
**Relation to prior work is very unclear**

**Rating:** 4
**Confidence:** 5

**Review:**

The author analyze the convergence properties of batch normalization for the ordinary least square (OLS) objective. They also provide experimental results on the OLS objective as well as small scale neural networks. First of all, understanding the properties of batch normalization is an important topic in the machine learning community so in that sense, contributions that tackle this problem are of interest for the community. However, this paper has a significant number of problems that need to be addressed before publication, perhaps the most important one being the overlap with prior work. Please address this point clearly in your rebuttal.

1) Overlap with Kolher et al. 2018: The authors erroneously state that Kolher et al. considered the convergence properties of BNGD on linear networks while after taking a close look at their analysis, they first derive an analysis for least-squares and then also provide an extension of their analysis to perceptrons. The major problem is that this paper does not correctly state the difference between their analysis and Kolher et al who already derived similar results for OLS. I will come back to this aspect multiple times below.

2) Properties of the minimizer
The authors should clearly state that Kolher et al. first proved that a^* and w^* have similar properties to Eq. 8. If I understand correctly, the difference seem to be that the algorithm analyzed in Kohler relies on the optimal a^* while the analysis presented here alternates between optimizing a and w. Is this correct? Is there any advantage in not using a^*? I think this would be worth clarifying.

3) Scaling property
I find this section confusing. Specifically,
a) The authors say they rely on this property in the proof but it is not very clear why this is beneficial. Can you please elaborate?
b) It seems to me this scaling property is also similar to the analysis of Kolher et al. who showed that the reparametrized OLS objective yields a Rayleigh quotient objective. Can you comment on this?
c) The idea of “restarting” is not clear to me, are you saying that one the magnitude of the vector w goes above a certain threshold, then one can rescale the vector therefore going back to what you called an equivalent representation? I don’t see why the text has to make this part so unclear. Looking at the proof of Theorem 3.3, this “property” seem to be used to simply rescale the a and w parameters.
d) The authors claim that “the scaling law (Proposition 3.2) should play a significant role” to extend the analysis to more general models. This requires further explanation, why would this help for say neural networks or other more complex models?

4) Convergence rate
It seems to me that the results obtained in this paper are weaker than previous known results, I would have liked to see a discussion of these results. Specifically,
a) Theorem 3.3 is an asymptotic convergence result so it is much weaker than the linear rate of convergence derived in Kolher et al. The authors require a sufficiently small step size. Looking at the analysis of Kolher et al., they show that the reparametrized OLS objective yields a Rayleigh quotient objective. Wouldn’t a constant step size also yield convergence in that case?
b) Proposition 3.4 also only provides a local convergence rate. The authors argue BNGD could have a faster convergence. This does seem to again be a weaker result. So again, I think it would be very beneficial if the authors could clearly state the differences with previous work.

5) Saddles for neural nets
The authors claim they “have not encountered convergence to saddles” for the experiments with neural networks. How did you check whether the limit point reached by BNGD was not a saddle point? This requires computing all the eigenvalues of the Hessian which is typically expensive. How was this done exactly?

6) Extension of the analysis to deep neural networks
The analysis provided in this paper only applies to OLS while Kolher et al. also derived an analysis for neural networks. Can the authors comment on extending their own analysis to neural nets and how this would differ from the one derived in Kolher et al.?

7) Experiments
How would you estimate the range of suitable step sizes (for both a and w) for BNGD for a neural network?

---

> ### Author Response · Authors · 2018-11-09
> **The key difference with prior works**
>
> Since most of the reviewer’s criticisms are based on comparisons with Kohler et al, we believe it is necessary to start with some overall comments on the differences between our independent work and theirs, and how these differences are significant in the attempt to gain insights on BNGD.
>
> The most important distinction is that in Kohler et al., the authors considered a modification of the BNGD on OLS in two ways: 1) The rescaling parameter is set to a value at every iteration to satisfy stationarity, instead of performing gradient descent; 2) the learning rate is chosen dynamically every step according to a rule which needs much more knowledge of the system. Thus, the analysis is fundamentally on a different algorithm from BNGD. In contrast, we analyse the original BNGD without these modifications, i.e. we consider a constant learning rate and also perform gradient descent on the rescaling parameters. We believe that this is an important distinction because the very goal of analyzing BNGD algorithm on a simplified model is to gain key insights into the *original* algorithm widely used in machine learning while circumventing the difficulties posed by a complex objective, say from a deep learning model.
>
> Consequently, none of the results we derived for the OLS model are weaker than, or can be deduced from the analysis presented in Kohler et al. In fact, we outline in some specific comments below (in next comment) that some of our results are stronger than those in Kohler et al if we take the simplifying modifications of the BNGD described above. In particular, the linear convergence of this modified BNGD follows directly from Eq.(13) of our analysis, which is a consequence of a simple projection argument (Eq. (26)). However, we must stress that we do not discuss these results explicit at length because we believe it is not relevant to our approach to study the original BNGD algorithm using simplified models, as we described above.

---

> > ### Comment · AnonReviewer1 · 2018-11-29
> > **Convergence results**
> >
> > First of all, I would like to apologize for my late reply. First, I would like to address what I think are the most important issues from the answer providing by the authors.
> >
> > a) I think it would be worth clarifying what you mean by “original BNGD”. To me, this includes two different algorithms: one is Batch normalization that perform a re-normalization, the other one is gradient descent which can be run using either constant or decreasing step-size. In practice, one would actually use **stochastic** gradient descent to optimize the weights of a neural network, for which you would need a decreasing step size to converge. I therefore do not quite agree with your statement that you are analyzing the BNGD "widely used today". If you want to make this claim, I think you should consider extending the analysis to stochastic gradients. That said, looking at Kohler et al., they do indeed require a learning rate that is chosen dynamically so in that sense, I would agree that what you analyze is closer to batch-norm.
> >
> > b) Thank you for providing clarifications regarding the analysis. I however still have some concerns regarding your analysis and still disagree that the results are stronger than prior work. One of the important complains I have is with Prop. 3.4 that you emphasized in your answer. There, you rely on linearizing the system close to a minimizer and therefore the bound derived in the proposition only holds locally. Your theorem does not say how large this neighborhood is so it is rather unclear what this result means in practice. I think this point should be made clear in the text.
> >
> > Minor comment on revised version: "linear networks (similar to the least-squares problem)"
> > It's very unclear what you mean by "similar" in the revised version, please clarify.

---

> > > ### Author Response · Authors · 2018-11-30
> > > **Convergence results**
> > >
> > > We thank the reviewer for the comments.
> > >
> > > a) We agree with the reviewer that we have not analyzed the stochastic optimization aspect of BN, but this was not claimed in our paper or replies --- BNGD refers to using GD to optimize the BN model, so in this sense we are analyzing the unchanged BNGD algorithm. Nevertheless, the stochastic aspect is interesting (and more challenging), and is an avenue for future work. We note that if we look at the proofs, the scaling property, the monotonicity  of weights and the phenomenon of BN decreasing the condition number compared with GD  remains true for the BNSGD case.
> > >
> > > b) Let us reiterate our convergence results:
> > > 1. If we take the same dynamic learning rate as in Kohler et al, from estimate eq.(13) (i.e. setting $\hat{\varepsilon}_k \equiv 1/L$) we **immediately** have linear convergence for **all** $k$. We chose not to write this statement explicitly in our results because 1) it is straightforward from estimate eq.(13) and 2) we are **not** analyzing the dynamic learning rate case.
> > > 2. If we take $\varepsilon_a = 1$, we have linear convergence even at constant learning rates (Th3.3(2) and Eq. (13)), but we do not have a rate constant estimate.
> > > 3. Our result prop. 3.4 shows something more: close to the optimum (the neighborhood can be estimated for $\delta<1$ according to eq.(15)), BNGD converges **faster** than GD on the original model. Prior work only established **same** convergence rate (even with dynamic learning rates) as GD, which again we can immediately deduce from estimate eq.(13).
> > >
> > > Point 1 shows our result implies prior OLS results if we take dynamic learning rates. Point 2 shows our result also establishes linear convergence for constant learning rates. Point 3 shows that our result shows a stronger convergence rate of BNGD over GD. This is not previously established. These justify our claim that our results are certainly stronger than previous results on OLS.

---

> > > > ### Comment · AnonReviewer1 · 2018-12-01
> > > > **Need clarification Prop 3.4**
> > > >
> > > > Thank you for the clarifications.
> > > >
> > > > a) ok, fair enough, thank you for clarifying this point.
> > > >
> > > > b)
> > > > 1. Regarding the linear rate, Theorem 3.3 is confusing to me, it seems to state a global convergence but I was unsure as to whether this is a linear rate or not. Could you please update the statement of the Theorem to clearly state it is a linear rate. I do not understand your statement "we do not have a rate constant estimate", can you please elaborate?
> > > >
> > > > 3. Can you provide more intuition behind the derivation of the Proposition? You seem to be excluding one dimension from the Hessian matrix and say for almost all u the hessian is better conditioned. Since you analyse a quadratic function, the curvature is however constant so I do not understand the reasoning in this proposition.

---

> > > > > ### Author Response · Authors · 2018-12-08
> > > > > **Clarification of Prop. 3.4**
> > > > >
> > > > > We thank the reviewer for the comments.
> > > > >
> > > > > 1. The convergence rate.
> > > > > Let us recall the convergence of GD on OLS: the convergence is linear and the rate constant $\rho$ depends on the step size $\varepsilon$ (suppose it is positive and less than $2/\lambda_{max}$). By contrast, we find the rate constant of BNGD depends on $\hat\varepsilon_k$ (see eq.(13) and (15)). If BNGD converges to a minimizer, then we have $\hat\varepsilon_k$ converges to a positive number $\hat\varepsilon$. However, we do not know whether $\hat\varepsilon$ is less than $2/\lambda_{max}$ or not.
> > > > > 1) If $\hat\varepsilon < 2/\lambda_{max}^*$, then we immediately have linear convergence. (A sufficient condition to this case is that $\varepsilon/||w_0||^2 < 2/\lambda_{max}$.)
> > > > > 2) If $\hat\varepsilon >= 2/\lambda_{max}^*$, then we have only sub-linear convergence.
> > > > > The latter is rare in our experiments on OLS.
> > > > >
> > > > > 3. Intuition of Prop.3.4
> > > > > The intuition here comes from the well-known Cauchy interlacing theorem. H^* can be regard as an orthogonal projection in the H-norm space. The proof of the eigenvalue property (Lemma A.1) is also similar to the proof of Cauchy interlacing theorem.

---

> ### Author Response · Authors · 2018-11-09
> **The detailed difference with prior works**
>
> Here we give the comments on the problems that the reviewer addressed one-by-one.
>
> 1) Difference between our analysis and Kohler et al.
>
> Here, for the linear networks, we meant to compare the OLS problem since this is the one we consider in this paper, thus we only mentioned the OLS part of Kohler’s analysis and its differences from ours (see opening comments and specific details furnished next). We acknowledge that analysis other than OLS is performed in Kohler et al, and we made this clear in the revision.
>
> 2) Properties of the minimizer
>
> As stated in the introductory comments, Kohler et al considers both an adaptive learning rate schedule and no explicit gradient descent on $a$ (rather a chosen to satisfy d(loss)/da=0 at each step). In our case, the learning rate is constant and gradient descent is performed on $a$. As stated before, this is an important distinction since we are studying the original BNGD algorithm, whereas Kohler et al’s OLS analysis (and also Ma & Klabjan, 17) is on a modified version of the BNGD algorithm. Note that we consider a version of a modified BNGD in the appendix B, where $a$ is also set according to d(loss)/da=0 (but learning rate is still constant). This tremendously simplifies the analysis but this is a different algorithm, and not the original BNGD that is widely used today.
>
> One practical advantage to performing GD on $a$ instead of setting it to satisfy stationarity is that it is hard to do in the general case where the model is not linear. It is not efficient to perform a line-search type augmentation (as suggested in Kohler et al) in practice, when the number of out-put features are high.
>
> In the revision, we have discussed this important distinction at some length in the second paragraph of the related work.
>
> 3) Scaling property
>
> a) First, the scaling property ensures that equivalent configurations (as in Definition 3.1) must converge or diverge together, with the same rate up to a constant multiple.
>
> This property is used in the proof of the main convergence result (Theorem 3.3). The sketch of the proof is as follows: we first prove that the norm of the parameters {|w_k|} is a monotone increasing sequence, thus either converges to a finite limit or diverges. The scaling property is then used to exclude the divergent case – if {|w_k|} diverges, then at some k the norm |w_k| should be large enough, and by the scaling property, it is equivalent to a case where |w_k|=1 and \varepsilon is small. But we can separately establish that for sufficiently small \varepsilon, the iterates converge from arbitrary initial condition |w_0|=1. This proves that {|w_k|} converges to a finite limit, from which the convergence of w_k and the loss can be established, after some work.
>
> b) The intuition of the scaling property follows from the Rayleigh quotient form, which we also consider in Appendix B, although this quotient form is not available for the original BNGD without the simplifications. This was not needed in Kohler et al. to prove their results, which are for small, dynamic learning rates. The scaling property is useful in controlling the stability of the system when large learning rates are explicitly allowed, as we have done in our work.
>
> c) idea of “restarting”
>
> Yes this is mostly right, except we emphasize that the rescaling is implicit, in the sense that equivalent configurations diverge or converge together. No actual rescaling happens and we are in fact analyzing the usual BNGD algorithm.
>
> We have reworded the sketch of the proof of Theorem 3.3 to make this more clear. In essence, the scaling property allows us to reduce the case of large |w_k| to the case of |w_k| = 1 and \varepsilon small, which we have established separately that it converges. This allows us to control the cases where learning rates are large and |w_k| grows in the early iterates.
>
> d) In Lemma A.9, we show the scaling property for a more general case, and in fact the same can be extended to general neural networks. Moreover, one can check that the monotonicity of {|w_k|} also holds generally. Hence, the same argument implies that in general, if one can establish convergence for |w_0|=1 and small \varepsilon, then one has the convergence of {|w_k|} for arbitrary learning rates, which will serve as a useful starting point to prove the convergence of {w_k}. This is work in progress.

---

> ### Author Response · Authors · 2018-11-09
> **The detailed difference with prior works (continued)**
>
> 4) Convergence rate
>
> a) For the “asymptotic” convergence rate, by asymptotic we meant that for large k, (e_k sufficiently small). These are *not* results for asymptotically small learning rates, but rather for arbitrary learning rates. Thus, it is incorrect to say that we require small step size for these results. In this sense, none of our results are weaker than previous results include those in Kohler et al, who actually requires a modification of the BNGD algorithm with special learning rate schedules. In our revision, we no longer call it asymptotic, as this creates the potential confusion that this result is asymptotic in \varepsilon.
>
> In fact, if one looks at Eq. (13) in our paper, if we modify the BNGD algorithm by setting the effective learning rate \hat{\varepsilon}_k to be 1/\lambda_{max} for every step, as is done in Kohler et al, then we would immediately obtain the same linear convergence rate (the convergence without this modification is much more difficult, and is the focus of the analysis in our paper). However, this requires the knowledge of \lambda_{max}, with which we can simply perform GD to get the same rate.
>
> In contrast, we obtain a better convergence rate in Prop. 3.4 if we are allowed to set \hat{\varepsilon}_k=1/\lambda_{max}, due to the better spectral property of H^*. This is a better rate than GD, even with knowledge of \lambda_{max}.
>
> Hence, our convergence results are stronger and not weaker than those in Kohler et al.
> We stress that although these rates can be obtained by modifying the BNGD algorithm, this is not the focus of this paper, where we consider the much harder case where \hat{\varepsilon}_k is not artificially set to be a constant value at every iteration. As discussed in the opening comments, this distinction is important in the philosophy of studying simple models: we need to preserve the algorithm.
>
> Finally, GD on the Rayleigh quotient objective is not shown to converge with arbitrary constant learning rates in Kohler et al, but we have showed this fact in Appendix B, where we also considered a similar simplifying modification. We do not discuss this in the main text because we would like to focus our discussion on the original BNGD, not a variant.
>
> b) This result is local only in the sense that it requires e_k to be sufficiently small, but this must hold for all k large enough since we already proved convergence. We note that if we took the simplifying assumption that \hat{\varepsilon}_k can be set to a constant, then we would have a bound for all k. Thus, this is again not a weaker result, and instead is a stronger one. See 4)a)
>
> 5) Saddles
>
> Here, we mean the OLS experiments. We will state it more clearly in revised paper.
>
> 6) Extension of the analysis to deep neural networks
>
> We have discussed our approach in extending this to general cases (including neural networks) in answer to 3)d). The basic difference in our approach again is that we would like to consider the original BNGD algorithm with potentially large, constant learning rates, as we believe this is the regime of interest for analysis.
>
> In this paper, we have done experiments in this case and discovered that the following insights translate to the general case (see Figure 3).
> (i) the BNGD allows large step sizes
> (ii) the optimal step size is insensitive
> (iii) the optimal performance is better than the optimal GD (one of the reasons is that BN decreased the condition number).
>
> However, the insights of Kohler et al. are totally different. Kohler et al. ascribed the acceleration of BN to the linear convergence (on specific models) while GD only achieves sublinear convergence. We think this argument is not convincing because GD’s sublinear convergence is for general objective functions (not for the special case they considered), and at worse it is sublinear, although many non-strongly-convex functions can achieve linear convergence using GD, for example, f(x,y) = x^2, or the BN on OLS.
>
> 7) Estimate the range of suitable step sizes.
>
> It is a good question. Generally, we have no analytical estimates. Heuristically speaking, the step size for w_k can be arbitrarily taken and the maximal step size for $a$ only depends on the neural network (independent on the datasets). Hence the suitable step size for $a$ for one dataset could be used to other datasets.

---

### Official Review · AnonReviewer3 · 2018-11-03
**Interesting but unsure analogies hold**

**Rating:** 4
**Confidence:** 3

**Review:**

The paper presents an analysis of the batch normalization idea on a simple OLS problem. The analysis is interesting as presented but several key questions remain, as described below. It is unclear that these questions are answered to the point where the insight gained can be considered transferable to BN in large Neural Network models.

- The reason why the auxiliary variable 'a' is included in the formulation (7) is unclear. The whole reason for using BN is to rescale intermediate outputs to have an expectation of zero and variance of one. The authors claim that BN produces "order 1" output and so 'a' is needed. Can you please explain his better?

- The scaling proposition 3.2 is claimed to be important, but the authors don't provide a clear explanation of why that is so. Two different settings of algorithms are presented where the iterates should roughly be in the same order if input parameters of the formulation or the algorithm are scaled in a specific way. It is unclear how this leads to the claimed insight that the BN algorithm is yielded to be insensitive to input parameters of step length etc. due to this proposition. also, where is the proof of this proposition? I couldn't find it in the appendix, and I apologize in advance if that's an oversight on my part.

- The 'u' referred to in eqn (14) is the optimal solution to the original OLS problem, so has form H^{-1} g for some g that depends on input parameters. Doesn't this simplify the expression in (!4)? Does this lead to some intuition on how the condition number of H^* relates to H? Does this operation knock off the highest or lowest eigenvalue of H to impact the condition number?

- Additionally, it is bad notation to use two-letter function names in a mathematical description, such as BN(z). This gets confusing very fast in theorems and proofs, though the CS community seems to be comfortable with this convention.

---

> ### Author Response · Authors · 2018-11-09
> **Insights of BNGD for large neural networks**
>
> In our paper, we have done experiments on some neural networks and discovered that the following insights is transferable (see Figure 3):
> (i) the BNGD allows large step sizes,
> (ii) the optimal step size is insensitive,
> (iii) the optimal performance is better than the optimal GD.
>
> For general large scale neural networks, it is much more difficult to strictly analyse those observations. However, some basic properties remain exactly. In Lemma A.9, we show the scaling property for a more general case, and in fact, the same can be extended to general neural networks where BNGD is used. Moreover, one can check that the monotonicity of {|w_k|} also holds generally. Hence, in general, if one can establish convergence for |w_0|=1 and small step size \varepsilon, then one has the convergence of {|w_k|} for arbitrary learning rates, which will serve as a useful starting point to prove the convergence of {w_k}.
>
> Below is our comments on the questions that the reviewer addressed.
>
> 1) The reason why the auxiliary variable 'a' is included.
>
> This is standard in the BN procedure and the reason of introducing `a` is explained in the original paper of BN (Ioffe and Szegedy, 2015). “Note that simply normalizing each input of a layer may change what the layer can represent.”. The variable `a` is there to make sure that the approximation power of the model remains the same.
>
> 2) The scaling property.
>
> First, the scaling property ensures that equivalent configurations (as in Definition 3.1) must converge or diverge together, with the same rate up to a constant multiple. Hence, this property is used in the proof of the main convergence result (Theorem 3.3). The sketch of the proof is as follows: we first prove that the norm of the parameters {|w_k|} is a monotone increasing sequence, thus either converges to a finite limit or diverges. The scaling property is then used to exclude the divergent case -- if {|w_k|} diverges, then at some k the norm |w_k| should be large enough, and by the scaling property, it is equivalent to a case where |w_k|=1 and \varepsilon is small. But we can separately establish that for sufficiently small \varepsilon, the iterates converge from arbitrary initial condition |w_0|=1. This proves that {|w_k|} converges to a finite limit, from which the convergence of w_k and the loss can be established, after some work.
>
> The insensitivity of choosing step size is stated in Proposition 3.5 and proved in Appendix A.5. It is not a direct consequence of the scaling property, but rather is related to the dynamics of the effective learning rate \hat{\varepsilon}, defined in (11).
>
> 3)  Condition number of H^*.
>
> The intuition here comes from the well-known Cauchy interlacing theorem. H^* can be regard as an orthogonal projection in the H-norm space. The proof of the eigenvalue property (Lemma A.1) is also similar to the proof of Cauchy interlacing theorem. H* does not knock out any eigenspace, but creates a zero-eigenvalue eigenspace with eigenvector u, thus it knocks out a bit from (in general) all eigenspaces and gives rise to the iterlacing property. This is the source of the acceleration.
>
> Using $Hu=g$, we can express equation (14) as $H^* = H - g^T g/(u^T g)$ which contains more letters.
>
> 4) We changed the notation ‘BN(z)’ to having BN on the subscript in revision.

---

### Official Review · AnonReviewer2 · 2018-11-03
**Interesting theoretical analysis for batch normalization**

**Rating:** 6
**Confidence:** 3

**Review:**

This paper provides a theoretical analysis for batch normalization with gradient descent (GDBN) under a simplified scenario, i.e., solving an ordinary least squares problem. The analysis shows that GDBN converges to a stationary point when the learning rate is less than or equal to 1, regardless of the condition number of the problem. Some practical experiments are carried out to justify their theoretical insights. The paper is in general easy to follow.

Pros:
This paper provides some insights for BN using the simplified model.
1. It shows that the optimal convergence rate of BN can be faster than vanilla GD.

2. It shows that GDBN doesn't diverge even if the learning rate for trainable parameters is very large.

Cons:
1. In the main theorem, when the learning rate for the rescaling parameter is less than or equal to 1, the algorithm is only proved to converge to a stationary point for OLS problem rather a global optimal.

2. To show convergence to the global optimal, the learning rate needs to be sufficiently small. But it is not specified how small it is.

Overall, I think this paper provides some preliminary analysis for BN, which should shed some lights for understanding BN. However, the model under analysis is very simplified and the theoretical results are still preliminary.

---

> ### Author Response · Authors · 2018-11-09
> **The convergence to global minimizers is proved in our revised paper**
>
> In our revised paper (Theorem 3.3), we give some cases where the convergence to global minimizer for arbitrary learning rates for the weights is proved. We stress that the small learning rates requirement is to avoid saddles, but it is not required for the stability of the algorithm. The smallness is quantified by Lemma A.14. Nevertheless, if we take Theorem 3.3 (2)’s condition, we do not need this requirement on the learning rates.
>
> We agree that we are analyzing BNGD on a specific problem, but we demonstrated through experiments that many insights hold generally. We believe it is appropriate to clearly analyze special and representative cases before attempting the general case, which is much more difficult.

---

### Meta-Review · Area_Chair1 · 2018-12-18
**Revise and resubmit**

**Confidence:** 4
**Recommendation:** Reject

**Metareview:**

The reviewers agree that providing more insights on why batch normalization work is an important topic of investigation, but they all raised several problems with the current submission which need to be addressed before publication. The AC thus proposes "revise and sesubmit".